# Discovery of non-retinoid compounds that suppress the pathogenic effects of misfolded rhodopsin in a mouse model of retinitis pigmentosa

Joseph T. Ortega[1], Jacklyn M. Gallagher[2,3], Andrew G. McKee[3], Yidan Tang[4], Miguel Carmena-Bargueňo[5], Maria Azam[1], Zaiddodine Pashandi[1], Marcin Golczak[1], Jens Meiler[4,6,7], Horacio Pérez-Sánchez[5], Jonathan P. Schlebach[2], Beata Jastrzebska[1]*

1 Department of Pharmacology and Cleveland Center for Membrane and Structural Biology, School of Medicine, Case Western Reserve University, Cleveland, Ohio, United States of America, 2 The James Tarpo Jr. and Margaret Tarpo Department of Chemistry, Purdue University, West Lafayette, Indiana, United States of America, 3 Department of Chemistry, Indiana University, Bloomington, Indiana, United States of America, 4 Department of Chemistry, Vanderbilt University, Nashville, Tennessee, United States of America, 5 Structural Bioinformatics and High-Performance Computing Research Group (BIO-HPC), UCAM Universidad Católica de Murcia, Guadalupe, Spain, 6 Center for Structural Biology, Vanderbilt University, Nashville, Tennessee, United States of America, 7 Institute for Drug Discovery, Leipzig University, Leipzig, Germany

* bxj27@case.edu

## Abstract

Pathogenic mutations that cause rhodopsin misfolding lead to a spectrum of currently untreatable blinding diseases collectively termed retinitis pigmentosa. Small molecules to correct rhodopsin misfolding are therefore urgently needed. In this study, we utilized virtual screening to search for drug-like molecules that bind to the orthosteric site of rod opsin and improve its folding and trafficking. We identified and validated the biological effects of 2 non-retinoid compounds with favorable pharmacological properties that cross the blood–retina barrier. These compounds reversibly bind to unliganded rod opsin, each with a $K_d$ comparable to 9-*cis*-retinal and improve opsin stability. By improving the internal protein structure network (PSN), these rod opsin ligands also enhanced the plasma membrane expression of total 36 of 123 tested clinical RP variants, including the most prevalent P23H variant. Importantly, these compounds protected retinas against light-induced degeneration in mice vulnerable to bright light injury and prolonged survival of photoreceptors in a retinitis pigmentosa mouse model for rod opsin misfolding.

## Introduction

Inherited mutations that disrupt protein folding are responsible for a variety of protein conformational diseases (PCDs) [1,2]. Membrane proteins, including G protein-coupled receptors (GPCRs), are especially prone to misfolding [3]. Misfolded membrane proteins that are detected by cellular quality control machinery are retained and degraded in the endoplasmic

**Data Availability Statement:** All relevant data are within the paper and its Supporting Information files.

**Funding:** This research was supported by grants from the National Institutes of Health (NIH) (R01EY032874 to B.J., R01GM129261, and R35GM152086 to J.P.S., and R01EY023948 to M. G.). This project was also supported by the NIH grant P30EY011373 awarded to the Visual Science Research Center Core at Case Western Reserve University and by the NIH P30CA043703 awarded to the Small Molecule Drug Discovery (SMDD) Core at Case Western Reserve University. Work performed in the Meiler laboratory was supported through the NIH grants awarded to J.M. (R01DA046138, R01HL122010, S10OD016216, S10OD020154, S10OD032234). J.M. received also funding from the Deutsche Forschungsgemeinschaft (DFG) through SFB1423 (421152132) and SPP 2363 (460865652). In addition, J.M. is supported by a Humboldt Professorship of the Alexander von Humboldt Foundation and by BMBF (Federal Ministry of Education and Research) through the Center for Scalable Data Analytics and Artificial Intelligence (ScaDS.AI). This work also was supported by a Sub-Award Agreement from the American Chemical Society with funds provided by a Prime Award from the Genentech Foundation awarded to J.M.G. Its contents are solely the responsibility of the authors and do not necessarily represent the official views of the name of the Genentech Foundation and the American Chemical Society. M. C.-B is a predoctoral student employed to the training of research staff financed by the Plan Propio de Investigación de la UCAM. This research was partially supported by the Plataforma Andaluza de Bioinformatica of the University of Malaga: Powered@NLHPC and by the supercomputing infrastructure of the NLHPC (ECM-02). The funders had no role in study design, data collection and analysis, decision to publish, or preparation of the manuscript.

**Competing interests:** The authors have declared that no competing interests exist.

**Abbreviations:** ADME, administration, distribution, metabolism, and elimination; AF, autofluorescence; AMD, age-related macular degeneration; BBB, blood–brain barrier; BRB, blood–retina barrier; DMSO, dimethylsulfoxide; ECL, extracellular loop; ER, endoplasmic reticulum; ERG, electroretinography; FACS, fluorescence-activated cell sorting; GI, gastrointestinal; GPCR, G protein-coupled receptor; HE, hematoxylin and eosin; HPLC, high-performance liquid chromatography; MD, molecular dynamics; MS, mass spectrometry;

reticulum (ER), which typically results in a loss of function [4,5]. One of the most important known PCDs is retinitis pigmentosa (RP), which is a blinding disease associated with mutations in the rod opsin (*RHO*) gene among other retina-specific genes [6,7]. Numerous mutations in *RHO* are known to enhance the misfolding of the nascent rhodopsin (Rho) protein in the ER, which compromises its trafficking, stability, and/or binding of its native 11-*cis*-retinal chromophore [7,8].

Pharmacological chaperones that target misfolded proteins and decrease folding energy barriers may correct protein misfolding and promote the proper routing, ameliorating the underlying mechanism of the disease [9,10]. The expression and trafficking of various misfolded Rho variants can be corrected by both retinoid and non-retinoid pharmacochaperones that bind and stabilize the unliganded opsin protein [11,12]. The 11-*cis*-retinal and its analog 9-*cis*-retinal are particularly effective at improving the folding and membrane targeting of pathogenic Rho mutants in vitro. However, their therapeutic utility is limited due to their sensitivity to light and chemical reactivity of retinal photo-metabolites. For these reasons, the development of pharmacological chaperones may require the discovery of non-retinoid compounds that bind to ligand-free opsin. Such chaperones may help suppress the degradation of destabilized rod opsin variants within the secretory pathway of rod cells where the supply of the endogenous stabilizing retinoids is limited.

Though several approaches have been previously employed to identify small molecules that rescue the expression of P23H and other misfolded Rho variants, most of these compounds have poor pharmacological properties or bioavailability, thus there are still no approved therapeutics for retinitis pigmentosa [11–13]. Computational-aided screening approaches offer an efficient means to identify novel small molecules that are capable of binding and stabilizing the folded opsin apoprotein. We recently employed these approaches to identify both a series of flavonoid and chromenone-containing compounds that bind to opsin's orthosteric site and partially correct the expression of certain *RHO* variants [4–16]. These compounds rescued the maturation and cellular trafficking of pathogenic mutants in cell culture and showed beneficial effects in vivo in the P23H Rho *knock-in* mouse model of RP. Though promising, many of the 100+ clinical Rho variants failed to respond to these compounds [16]. Therefore, the discovery of pharmacological chaperones with favorable pharmacological properties that correct a wider array of RP variants is still needed [16]. Such efforts may also provide avenues for the development of targeted, variant-specific combination therapies such as those that are currently offered for the treatment of cystic fibrosis [17]. The extent to which various non-retinoid pharmacochaperones could potentially rescue distinct classes of misfolded *RHO* variants remains unclear.

In this study, we utilized virtual screening to identify alternative compounds that bind within the orthosteric site and stabilize the opsin protein. We then employed in vitro biochemical assays, cellular measurements, and animal model studies to validate the pharmacological utility of our top drug candidates. We identified 2 compounds that bind to rod opsin in vitro, enhance its stability, and restore its maturation and plasma membrane expression in a model cell line stably expressing the most common P23H Rho mutant. Using deep mutational scanning, we compared the effects of these compounds on the expression of 123 known clinically relevant *RHO* variants and showed that these compounds appear to be effective against multiple variants that could be effectively corrected by 9-*cis*-retinal, and a subset of variants that are distinct from those the most responsive to retinoids [8]. Moreover, we identified several mutants that differentially respond to our 2 hit compounds. Structural models of selected variants bound to these compounds suggest that the variant-specific effects arise from subtle differences in the binding orientation. MD simulations and protein structure network (PSN) analysis showed that these compounds reduce structural fluctuations caused by mutations and

ONH, optic nerve head; ONL, outer nuclear layer; PBS, phosphate buffer saline; PCD, protein conformational disease; PME, particle mesh Ewald; PNA, peanut agglutinin; PSN, protein-structure network; PVDF, polyvinylidene difluoride; RMSD, root mean square deviation; ROS, rod outer segment; RP, retinitis pigmentosa; SAR, structure-activity relationship; SD, standard deviation; SD-OCT, spectral domain-optical coherence tomography; SLO, scanning laser ophthalmoscopy; SPC, simple point charge; UPR, unfolded protein response; WT, wild-type.

improve internal residue–residue interactions shifting the receptor's conformation towards WT-like. Importantly, we showed that both hit compounds accumulate within the eyes of mice for several hours after their intraperitoneal (i.p.) administration. We validated the therapeutic potential of these compounds in 2 mouse models of retina degeneration, *Abca4$^{-/-}$Rdh8$^{-/-}$* mice, a model of acute light damage, and in an RP model, P23H Rho *knock-in* mice. Both compounds protected photoreceptors from cell death induced by light and the pathogenic P23H *RHO* mutation. Together, our findings suggest that the molecules identified in this study represent promising lead compounds for the development of precision therapeutics for RP and other visual retinopathies.

## Results

### In silico discovery of new non-retinoid ligands of rod opsin

We recently found that dietary flavonoids and unrelated chromenone-containing compounds bind within the orthosteric site of rod opsin and act as pharmacological chaperones for RP-linked mutants [14–16]. To expand on the growing list of Rho pharmacophores, we carried out a virtual screen of a library of non-retinoid small molecules commercially available within the *Zinc* database (http://zinc.docking.org) [18]. We performed pharmacophore-based screening using as a molecular descriptor the rod opsin ligand pharmacophores that we recently discovered as well as those of other previously reported non-retinoid opsin ligands [13,14,16]. Comparative docking of these compounds within the crystal structure of unliganded bovine rod opsin (PDB ID: 3CAP) revealed several compounds with predicted binding free energies that were the same or lower to that of quercetin (−9.3 kcal/mol) [14]. From this subset of compounds, we selected 2 commercially available compounds that we will refer to hereafter as JC3 and JC4 (**Figs 1** and **2**). These compounds form a network of favorable interactions in proximity to K296 residue within the orthosteric site of rod opsin with binding free energies of −9.5 kcal/mol and −9.7 kcal/mol, respectively (**Fig 2A**). Both compounds formed direct interactions with various residues that are known to stabilize the natural 11-*cis*-retinal chromophore within the binding pocket including Ala117, Ala292, Tyr191, and Glu181 (**Fig 2B**). The most favorable orientations of these ligands also placed them near Trp265 and Tyr268, similarly to 11-*cis*-retinal.

JC3 and JC4 were also chosen for experimental characterization due to their favorable drug-like properties according to the analysis of their absorption, distribution, metabolism,

| Compound | Structure | Chemical Name | PubChem ID | Opsin Binding Free Energy (kcal/mol) |
|---|---|---|---|---|
| JC3 | | 2-(1,3-benzodioxol-5-yl)-3-(4-methylphenyl)-1,3-thiazolidin-4-one | 2860307 | -9.5 |
| JC4 | | 2-(4-chlorophenyl)-3-methyl-1-(4-methylphenyl)-2H-pyrrol-5-one | 12006040 | -9.7 |

**Fig 1. Chemical compound characterization.**

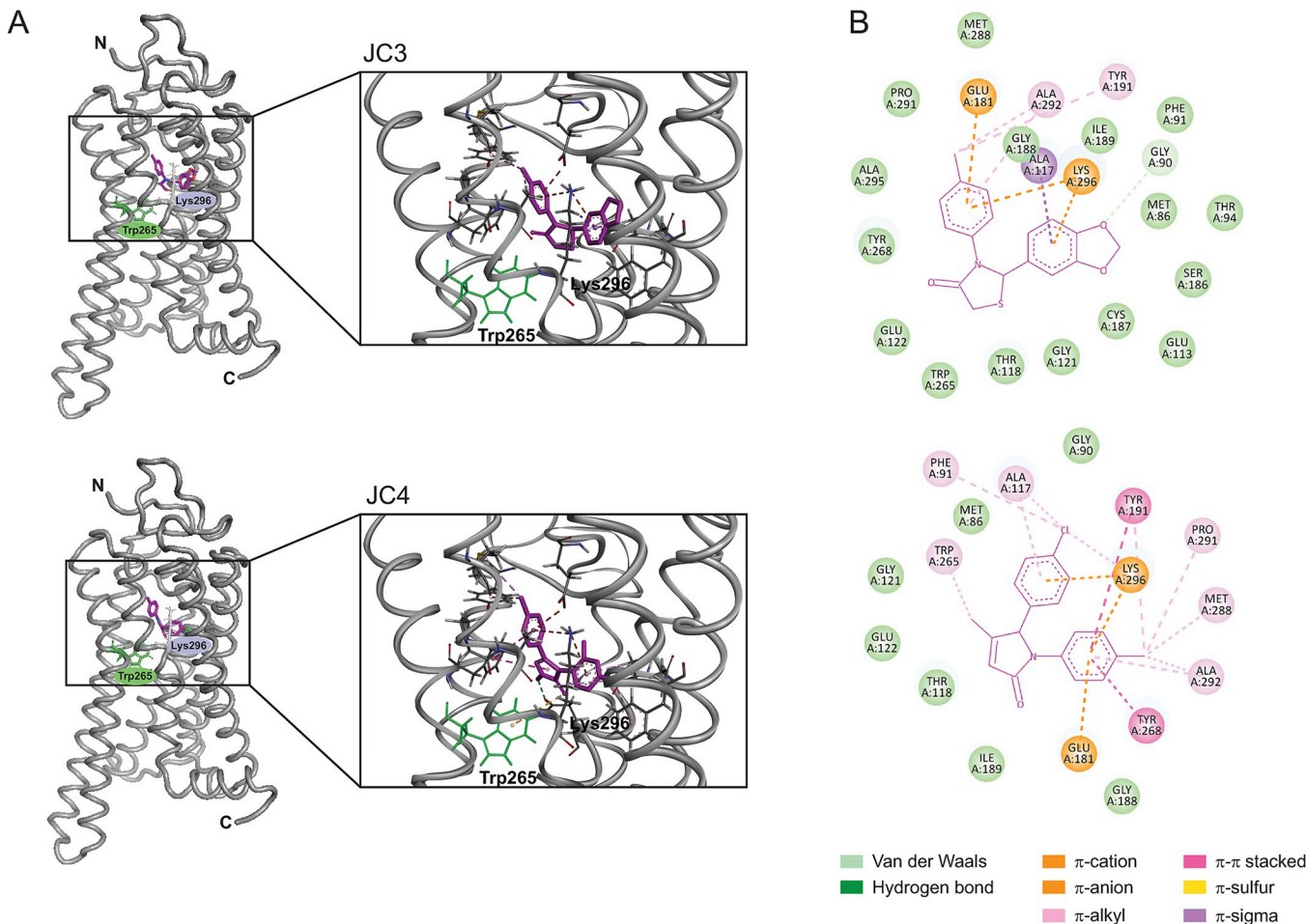

**Fig 2. Molecular docking of the compounds to opsin's orthosteric site.** (A) Two compounds JC3 and JC4 were identified in silico. The structures of bovine rod opsin (PDB ID: 3CAP) with the best binding poses of these compounds docked to the orthosteric binding pocket and close-ups of the binding site are shown. The ribbons in the rod opsin structure are shown in gray. The compounds are shown in purple. Trp265 is shown as green sticks. (B) Two-dimensional representation of the low energy structures obtained from molecular docking simulations. The interactions between the compound and the residues in the opsin's polypeptide chain are shown. The structures obtained from molecular docking simulations (A) and the two-dimensional representations (B) were visualized with the Biovia Discovery Studio Visualizer 17.2.0 visualizer.

and elimination (ADME) profiles, which were determined using SWISSADME tools (http://www.swissadme.ch) [19]. These tools score compounds based on their predicted lipophilicity, hydrophilicity, solubility, absorption in the gastrointestinal (GI) tract, and permeability to the blood–brain barrier (BBB). The main ADME parameters are presented in **Table 1**. None of the selected compounds violated Lipinski's rule of 5 [20]. The JC3 and JC4 compounds are

**Table 1. ADME parameters.**

| Molecule | MW (g/mol) | ESOL log S | ESOL class | Ali log S | GI absorption | BBB permeant | Lipinski #violations | PAINS #alerts | Synthetic accessibility |
|---|---|---|---|---|---|---|---|---|---|
| JC3 | 313.37 | −4.28 | Moderately soluble | −4.57 | High | Yes | 0 | 0 | 3.3 |
| JC4 | 297.78 | −4.49 | Moderately soluble | −4.12 | High | Yes | 0 | 0 | 2.94 |

ADME, administration, distribution, metabolism, and elimination; BBB, blood–brain barrier; GI, gastrointestinal.

moderately soluble and have predicted high absorption in the GI tract. Importantly, these results suggest that JC3 and JC4 should be able to cross the BBB and thus likely the blood–retina barrier (BRB). These properties are critical considerations for the discovery of orally bioavailable small molecules that may correct the folding of Rho within the eye.

## Impact of JC3 and JC4 on pigment regeneration and stability in vitro

To confirm that JC3 and JC4 bind within the orthosteric binding pocket of rod opsin, we used a previously established Trp fluorescence quenching assay to track ligand binding [14,16]. Incubation of purified rod opsin membranes (ROS) with increasing concentrations of these compounds resulted in a progressive increase in the quenching of intrinsic Trp fluorescence of opsin at 330 nm (**Fig 3A**). This change results from the conformational rearrangement of Trp265 within the ligand-binding pocket upon ligand binding [21]. Fits of the observed quenching data with a single-site binding model suggest equilibrium dissociation constant ($K_d$) values for JC3 and JC4 of $175 \pm 20$ nM and $98.5 \pm 33$ nM, respectively. Consistent with in silico docking scores, these findings indicate that both molecules bind within the orthosteric pocket and that JC4 binds with higher affinity relative to JC3.

Next, we examined the effect of JC3 and JC4 binding on the ability of rod opsin to bind 9-*cis*-retinal and regenerate its isochromophore. The binding of 9-*cis*-retinal results in the appearance of an absorption peak at 485 nm, a well-known spectroscopic signature for the formation of a Schiff base between the retinal and Lys296 in the protein backbone [22]. As expected, pretreatment of opsin membranes with JC3 and JC4 did not generate a pigment. However, pre-saturation of the orthosteric pocket with these compounds partially decreased the binding efficiency of the retinal chromophore following a 10-min incubation with 9-*cis*-retinal (**Fig 3B**). This inhibition is concentration dependent and more pronounced at 100 μM relative to 10 μM. Pre-saturation with JC3 and JC4 also slowed the regeneration kinetics, the regeneration half-time of isorhodopsin (isoRho) increased from $3.9 \pm 0.6$ min to $8.3 \pm 0.9$ min and $8.4 \pm 1.4$ min in the presence of 100 μM JC3 and 100 μM JC4, respectively (**Fig 3C**). Together, these observations confirm that JC3 and JC4 compete with retinal for binding of the rod opsin orthosteric site, but do not inhibit the Schiff base formation between retinal and the receptor.

The binding of pharmacochaperones typically enhances the stability of natively folded proteins [14]. To determine whether the binding of these compounds stabilizes opsin, we incubated opsin ROS membranes with each JC compound and then compared the melting temperature ($T_m$) of the bound protein using a thermal shift assay. The $T_m$ of opsin increased slightly in the presence of 0.1 μM JC3 ($56.5 \pm 0.3°C$) or 0.1 μM JC4 ($57.5 \pm 0.7°C$) relative to untreated opsin under these conditions ($55.4 \pm 0.4°C$) (**Fig 3D**). As expected, this stabilization by JC3 or JC4 was not observed if the pigment was first regenerated by retinal, which would inhibit JC3 and JC4 binding. IsoRho and the pigments regenerated in the presence of JC3 and JC4 all exhibited comparable kinetic stability at 55°C, which indicates that the binding of these compounds does not impair the stability of the mature pigments (**Fig 3E**).

## Pharmacological activity of JC3 and JC4

To examine the effect of JC3 and JC4 compounds on the function of the visual receptor, we measured their effects on the receptor-mediated photoactivation of G-protein transducin ($G_t$). Briefly, opsin ROS membranes were first preincubated with JC3 or JC4, followed by the regeneration with 9-*cis*-retinal, and with purified $G_t$ protein, and then Trp fluorescence was used to monitor changes in Trp fluorescence of $G_{t\alpha}$ occurring due to the exchange of GTPγS and the dissociation of the receptor-$G_t$ complex upon photoactivation (**Fig 4A and 4B**). The initial

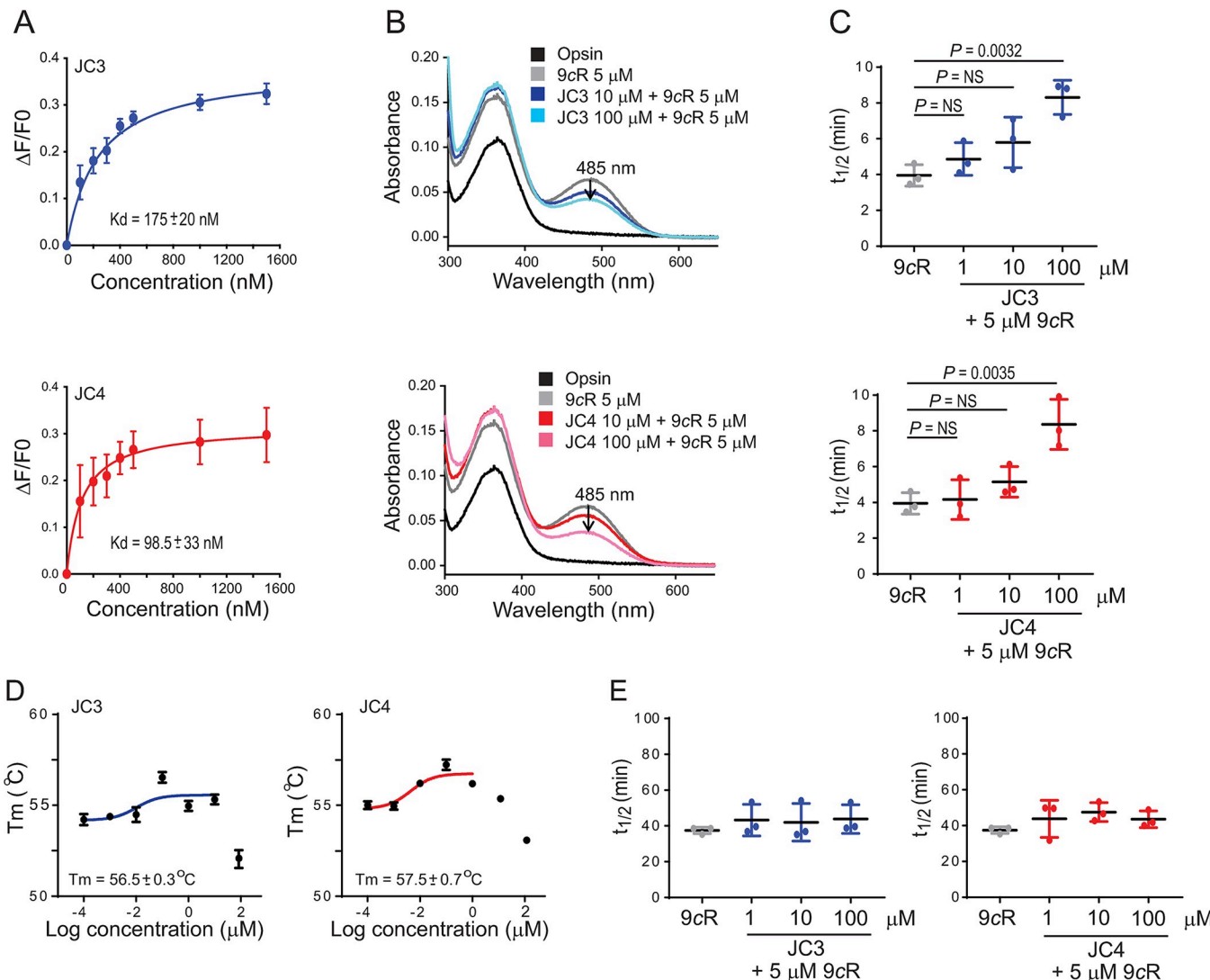

**Fig 3. Binding of JC3 and JC4 to rod opsin, their effects on pigment regeneration, stability, and signaling.** (A) The binding of the identified compounds was determined by quenching the opsin's intrinsic Trp fluorescence. Compounds were added to opsin membranes at different concentrations (100–1,500 nM) and changes in the fluorescence intensity at 330 nm were recorded and plotted as a function of the compound concentration. The binding curves were fitted using PRISM GraphPad 7.02 software. The $K_d$ values of each compound were calculated and averaged from triplicates. These values ± SDs are shown in the figure. (B) The UV-visible spectra of isoRho regenerated with 5 μM 9-*cis*-retinal (9*c*R) after treatment of opsin membranes with the JC3 and JC4 compounds at 10 μM and 100 μM concentrations. (C) The half-life ($t_{1/2}$) of isoRho regenerated with 5 μM 9*c*R and upon the treatment of opsin membranes with the new compounds at 1, 10, and 100 μM concentrations. (D) The effect of JC3 and JC4 on rod opsin stability. The temperature of melting ($T_m$) was determined for the opsin membrane in the presence of the JC compounds at different concentrations (0.0001–100 μM) by using a fluorescent probe BFC. The samples were incubated in the step-wise temperature gradient up to 99.9˚C. The obtained values of fluorescence were plotted as a function of temperature and the melting temperature was calculated using GraphPad 7.02 software. The $T_m$ for each compound is presented as a function of compound concentration and the values obtained at 0.1 μM compound concentration are shown in the figure. Error bars represent SD. Each measurement was repeated 3 times. Statistical analysis was performed with the one-way ANOVA and Turkey post hoc tests. The statistically different changes (*P*) are indicated in the figure. NS, not statistically significant. (E) The half-time ($T_{1/2}$) of isoRho decay at 55˚C. IsoRho was regenerated with 5 μm 9*c*R upon the treatment of opsin membranes with JC3 or JC4 compounds at 1, 10, and 100 μM concentrations. The numerical data can be found in S1 Data.

rates of $G_t$ activation decreased slightly relative to the untreated control ($k_{initial}$ = $5.1 \pm 0.3 \times 10^{-3}$ s$^{-1}$) when opsin membranes were preincubated with 10 μM JC3 ($k_{initial}$ = $4.4 \pm 0.3 \times 10^{-3}$ s$^{-1}$) or JC4 ($k_{initial}$ = $4.3 \pm 0.4 \times 10^{-3}$ s$^{-1}$). To determine whether this result coincides with the inhibition of cellular signaling, we evaluated the effect of JC3 and JC4 on

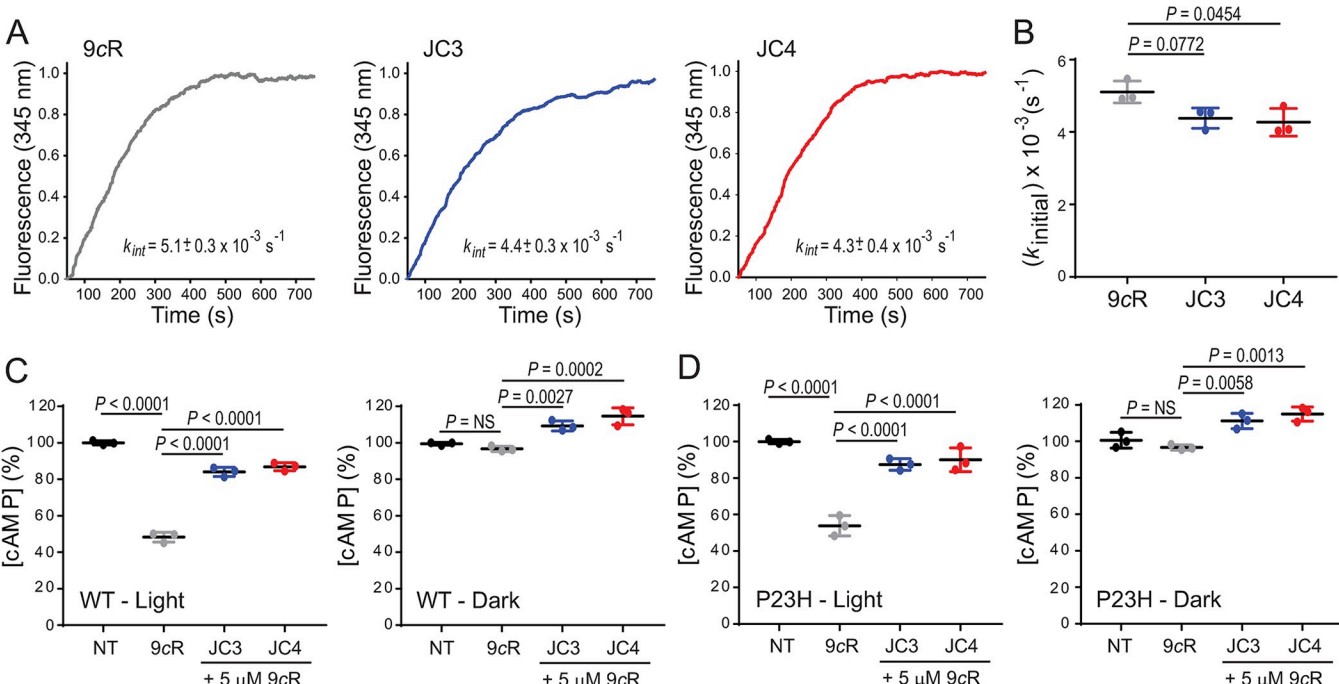

**Fig 4. The effect of JC3 and JC4 on the $G_t$ activation rates.** (A) The rod opsin membranes were incubated with a 10 μM compound prior to pigment regeneration with 5 μM 9-*cis*-retinal (9*c*R). The activity of illuminated isoRho was recorded by monitoring changes in the Trp fluorescence at 345 nm upon the addition of 10 μM GTPγS. These changes related to the dissociation of $G_{t\alpha}$ were plotted as a function of time. The representative plot is shown. The excitation and emission wavelengths were 295 nm and 345 nm, respectively. (B) The initial rates and error bars (SD) calculated for each condition plotted are shown. Each measurement was performed 3 times and the experiment was repeated. (C and D) The effect of JC3 and JC4 on the function of WT and P23H isoRho in cultured cells, respectively. In cultured cells, Rho can signal through $G_i$ signaling. Changes in the levels of cAMP upon light stimulation were monitored in the NIH-3T3 cells stably expressing WT (C) or P23H rod opsin (D). These cells were regenerated with 5 μM 9-*cis*-retinal (9*c*R) or incubated with JC3 or JC4 prior to regeneration. Levels of cAMP obtained in the illuminated cells were compared to the levels in cells kept in the dark. The cells were treated with the compound for 16 h before isoRho regeneration for 2 h. Cells treated with 9*c*R only and non-treated (NT) cells were used as controls. Each condition was performed in triplicate and the experiment was repeated. The one-way ANOVA and Turkey post hoc tests were used for the statistical analysis. The statistically different changes (*P*) are indicated in the figure. The numerical data can be found in S1 Data.

signaling in situ in cultured NIH-3T3 cells heterologously expressing WT rod opsin and compared it to the P23H rod opsin mutant. In the absence of $G_t$ Rho signals through $G_i$, resulting in a decrease in cellular levels of cAMP [23]. Thus, we examined the level of cAMP in cells treated with the JC compounds followed by regeneration of isoRho and compared it with cells incubated with 9-*cis*-retinal only, either after their exposure to light or kept in the dark (**Fig 4C and 4D**). Upon light stimulation, in the cells regenerated with 9-*cis*-retinal the cAMP concentration lowered to approximately 50% of the level present in the non-regenerated cells in both, cells expressing WT and P23H mutant receptors. However, in the cells preincubated with JC3 or JC4, the decrease in cAMP level was much lower and its concentration reached about 80% to 90% as compared to non-regenerated cells, indicating a partial antagonistic effect of JC3 and JC4 compounds. In addition, cells that were kept in the dark upon treatment with JC3 and JC4 showed a slight increase, in the cAMP levels as compared to non-treated cells. Although, the reason for such a result is unclear it could be related to the silencing of constitutive opsin activity by the JC compounds. These results indicate that JC3 and JC4 compounds act as new rod opsin ligands with partial antagonist activity. Altogether, our computational and in vitro receptor binding analyses, along with the functional assays demonstrate that JC3 and JC4 could modulate Rho signaling through direct interaction with the rod opsin protein.

## Impact of JC3 and JC4 on the plasma membrane expression of P23H rod opsin

The P23H mutation in Rho causes misfolding, ER retention, and enhanced degradation in a manner that causes degeneration of the retina. The proper cellular trafficking of Rho is critical for its function. Thus, we examined the effect of JC3 and JC4 on membrane trafficking of P23H rod opsin in the NIH-3T3 cells stably expressing this receptor. Neither compound causes appreciable cellular toxicity at 10 μm concentration in the 4 cell lines tested, which included NIH-3T3 and HEK-293 cells as well as the retina-derived cell lines 661W and ARPE19 (**Fig 5A**). Interestingly, treatment with JC3 and JC4 at 10 μm for 16 h enhanced the surface immunostaining of P23H rod opsin relative to untreated cells, with JC4 having a slightly higher efficacy (**Fig 5B and 5C**). Importantly, the effect generated by JC4 was comparable to that of 9-*cis*-retinal. These findings demonstrate that JC compounds improve the plasma membrane expression of the most common RP-linked rod opsin mutant in the absence of a retinal chromophore.

To examine whether the enhanced membrane localization of P23H rod opsin arises from changes in the total expression or differences in maturation, we utilized western blotting to compare the effects of these compounds on the glycosylation state of rod opsin. Wild-type (WT) rod opsin expressed in the NIH-3T3 cells predominantly bears higher weight, mature glycans, and migrates at an apparent molecular weight of 55 kDa (**Fig 5D**). By comparison, the predominant form of P23H rod opsin has an apparent molecular weight of 37 kDa, which reflects its impaired maturation within the secretory pathway (**Fig 5D**). Treatment with 9-*cis*-retinal stabilized P23H rod opsin in a manner that resulted in the appearance of both higher weight mature P23H glycoforms and/ or oligomers. This interpretation of the observed protein migration pattern is supported by the decreased apparent molecular weight of the P23H protein following treatment with the PNGaseF glycosidase. An increase in the relative abundance of the mature P23H glycoform was also observed in response to treatment with JC4 and, to a lesser extent, upon treatment with JC3 (the ratio of the mature to immature receptor increased to approximately 1.3 ± 0.2 and 1.6 ± 0.2 fold in the presence of JC3 and JC4, respectively) (**Fig 5E**). Similar results showing an increase in the plasma membrane localization of P23H rod opsin upon treatment with the JC compounds were obtained in the photoreceptor-derived 661W cells stably expressing this mutant receptor. Interestingly, the ratio of mature to immature receptors was slightly greater in these cells and reached 1.5 ± 0.2 fold in the presence of JC3 and 3.0 ± 0.5 fold in the presence of JC4 (**S1 Fig**). Together, these observations show that JC3 and JC4 enhance the maturation of the nascent P23H rod opsin protein within the ER in a manner that ultimately increases its plasma membrane expression. In addition, these findings stimulated a focused analysis of the effects of the JC3 and JC4 compounds on the processing of various RP mutants.

## Survey of the mutation-specific effects of JC3 and JC4

Though several molecules that partially correct the misfolding of P23H have been previously reported, the effects of these compounds vary tremendously in their effects against the full spectrum of pathogenic *RHO* variants [8,16]. To survey the effects of JC3 and JC4 against the large spectrum of known clinically relevant variants, we utilized deep mutational scanning to quantitatively compare their impacts on the plasma membrane expression of 123 clinical variants in HEK-293T cells as previously described [8,16]. Treatment of cells with JC3 and JC4 increased the expression of 32 and 26 of the 123 tested rod opsin variants, respectively, by at least 5% (**Fig 6A and 6B**). Importantly, both compounds corrected membrane trafficking of the most prevalent *RHO* variant P23H [6]. Though these compounds generally enhance the

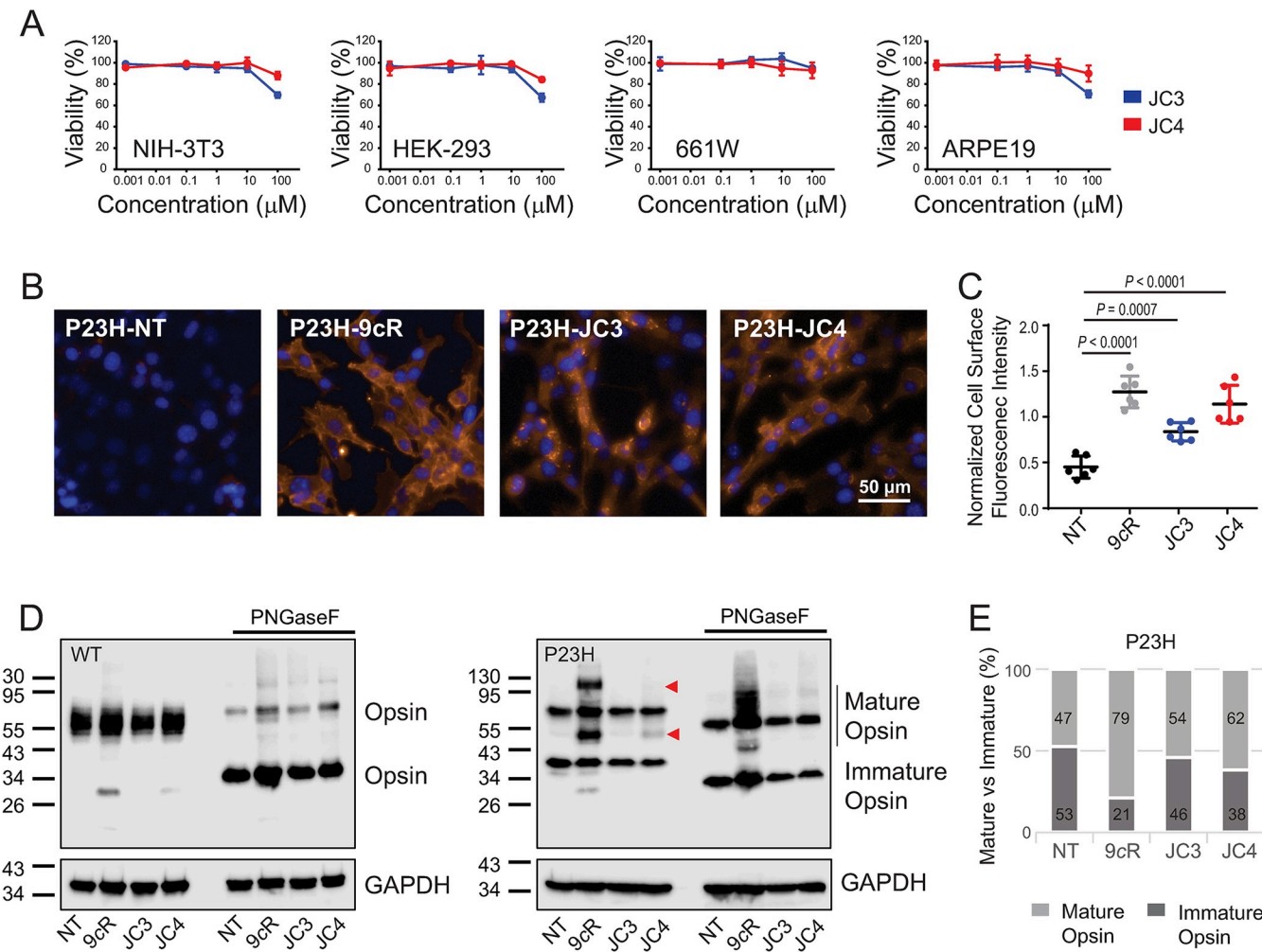

**Fig 5. Effect of JC3 and JC4 on opsin membrane targeting.** (A) Cytotoxicity of JC3 and JC4 compounds. Toxicity of the identified compounds was tested in 4 cell lines, the NIH-3T3, HEK-293 cells, photoreceptor-derived 661W cells, and retinal pigment epithelium (RPE)-derived ARPE19 cells by MTT assay. The cells were treated with the compounds at 0.001–100 μM concentration for 24 h. The results are shown as a percentage of viable cells compared to non-treated control cells. (B) The fluorescence images of the NIH-3T3 cells stably expressing P23H rod opsin treated with the JC compounds at a final concentration of 10 μM or 5 μM 9-*cis*-retinal (9*c*R) for 16 h. Each condition was performed in triplicate and the experiment was repeated. The next day, cells were immunostained with the anti-Rho monoclonal antibody recognizing the N-terminal epitope of this receptor and the Alexa-Fluor 594-conjugated anti-mouse secondary antibody (orange) to detect the cell surface expression. The nuclei of the cells were labeled with DAPI (blue). The images were taken with a high-content imaging operetta microscope at 20× magnification. Scale bar, 50 μm. (C) Quantification of the cell surface fluorescence intensity. Statistical analysis was performed with the one-way ANOVA and Turkey post hoc tests. The statistically different changes (*P*) are indicated in the figure. (D) Immunoblot showing the effect of 2 compounds JC3 and JC4 on the expression level of WT and P23H rod opsin in the NIH-3T3 cells stably expressing these receptors. Total cell extracts (50 μg) were loaded and separated using SDS-PAGE gel, followed by transfer to polyvinyl difluoride membrane (PVDF). Rod opsin was detected with the 1D4 anti-Rho antibody detecting the C-terminal epitope. GAPDH was detected with an anti-GAPDH antibody and used as a loading control. PNGaseF-treated samples were deglycosylated for 1 h at room temperature prior to loading onto the gel. The experiment was repeated 3 times. Representative immunoblots are shown. (E) Quantification of band intensities of mature and immature P23H rod opsin in non-treated cells, treated with 9-*cis*-retinal and 2 JC compounds in the blots shown in D. Raw images of triplicate immunoblots can be found in S1 Raw Images. The numerical data can be found in S1 Data.

expression of a common subset of these variants, the magnitude of their responses varied considerably. For instance, M39R, L47R, L57R, R135L, A164E, P180A, D190N, and ΔC264 exhibited larger gains in expression in the presence of JC3 relative to JC4. In contrast, A169P, H211R, and S297R preferentially responded to JC4 over JC3. Notably, many of the destabilizing mutations that can be partially corrected by these compounds are located within the N-terminus or within the extracellular loop (ECL) 2 region that forms the plug stabilizing retinal

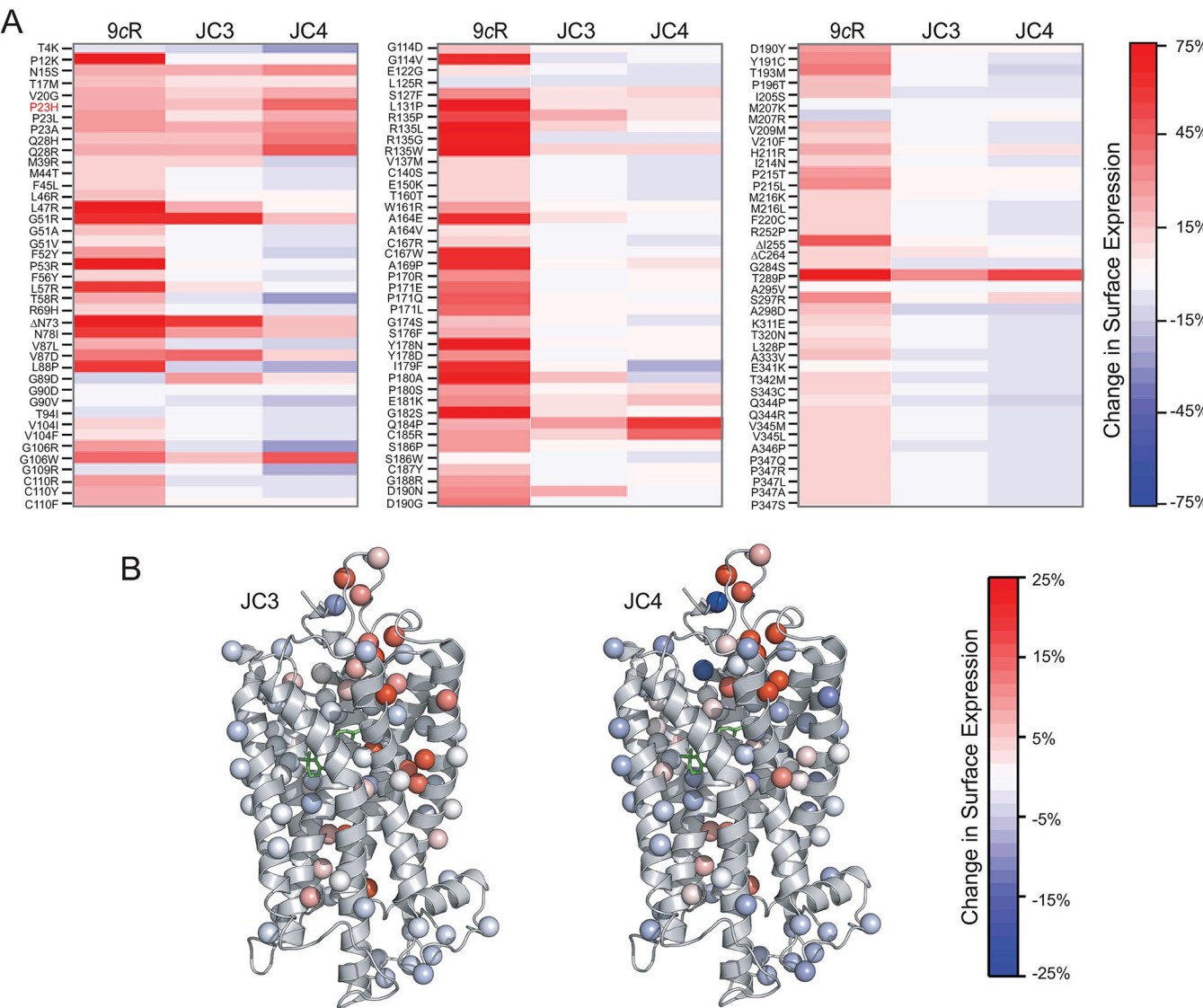

**Fig 6. Effect of JC3 and JC4 on membrane expression of RP-linked rod opsin mutants.** Deep mutational scanning was used to examine the chaperone effect of JC3 and JC4 on multiple RP-linked rod opsin mutants (see Experimental procedures). The change in the plasma membrane levels of these variants upon treatment with a 10 μM compound was quantified and compared to the treatment with 5 μM 9-*cis*-retinal (9*c*R). (A) The cell surface immunostaining intensities for a collection of RP variants bearing individual amino acid substitutions examined in HEK-293T cells were depicted as heatmaps. The values represent the average from 2 biological replicates, and color bars indicate the scale of the observed effects under each condition. Red indicates an increase and blue a decrease in plasma membrane expression under each condition. (B) The Cα of mutated side chains are rendered as spheres in the context of the 3D structure of Rho (PDB ID: 1U19). Spheres are colored according to the average change in the plasma membrane expression in the presence of JC3 and JC4. The 11-*cis*-retinal chromophore is shown as green sticks to visualize the chromophore binding region. The numerical data can be found in S1 Data.

within the binding pocket. Other sensitive mutations are located in the proximity of the retinal within the orthosteric site. A few residues are located within transmembrane helix (TM) 1 and are possibly important for receptor–receptor interactions required for proper folding and trafficking. Though the effects of these compounds are modest relative to that of the 9-*cis*-retinal isochromophore for some mutants, certain variants do appear to be more sensitive to the JC compounds. P23H, P23A, Q28H, Q28R, Q184P, C185R are more sensitive to JC4 while G89D exhibits sensitivity to JC3 but does not respond to 9-*cis*-retinal. Together, these results revealed the scope of the mutation-specific effects of JC3 and JC4 across the spectrum of known RP-

related rod opsin mutants. Consistent with our previous findings [8,16], these results suggest that effective pharmacochaperones for Rho misfolding will eventually need to be targeted to a specific subset of clinical variants to ensure the success of the treatment.

## Impacts of JC3 and JC4 on retinal degeneration

To validate the therapeutic potential of JC3 and JC4 in vivo, we examined their effectiveness in 2 mouse models of retina degeneration $Abca4^{-/-}Rdh8^{-/-}$ and $Rho^{P23H/+}$ knock-in mice. Accumulation of ligand-free opsin after photobleaching in the retina of $Abca4^{-/-}Rdh8^{-/-}$ mice accelerates the degeneration of their photoreceptors as a result of constitutive signaling activation [24–26]. Retinal health is compromised in these mice due to delayed clearance of all-*trans*-retinal photoproducts and/or regeneration of functional Rho. Given that JC3 and JC4 stabilize opsin in vitro, we assessed whether treatment with these compounds protects their retina from light insult. $Abca4^{-/-}Rdh8^{-/-}$ mice were treated with an i.p. injection of 100 mg/kg body weight (b.w.) of JC3 or JC4 30 min prior to a 30-min exposure to 10,000 lux light. For comparison, we included control mice that were either withheld from treatment and kept in the dark or treated with a vehicle and subjected to the same illumination. The effects of each compound on both retinal morphology and visual function were then examined 7 days later. In vivo retina imaging (spectral domain optical coherence tomography (SD-OCT) and scanning laser ophthalmoscopy (SLO)) and histological analysis showed that, while vehicle-treated mice exhibited a thinning of their outer nuclear layer (ONL), the retina of the JC3 and JC4-treated mice closely resembled those of the non-treated dark-adapted mice (**Fig 7A–7C**). In addition, the exposure of $Abca4^{-/-}Rdh8^{-/-}$ mice to bright light activates the migration of microglial cells to the retina to clear dying photoreceptors as evidenced by the appearance of autofluorescence (AF) spots [26,27]. Though vehicle-treated mice exhibited an increase in the number of AF spots, this pathology was not detected in mice treated with either JC3 or JC4 (**Fig 7D and 7E**). We next examined retinal function by measuring the electroretinography (ERG) responses, which provide a measure of retina activity in response to a light stimulus. Both dark-adapted ERG responses of rod photoreceptors and light-adapted ERG responses of cone photoreceptors were much smaller for mice treated with a vehicle and exposed to bright light relative to mice that were not subjected to illumination (**Fig 7F**). However, treatment with either JC3 or JC4 preserved normal retina function. By comparison, ERG responses for the treated mice closely resembled those of unexposed control mice (**Fig 7F**). Together, our results indicate that JC3 and JC4 protect the retina of $Abca4^{-/-}Rdh8^{-/-}$ mice from retinal degeneration caused by bright light injury.

We next examined whether JC3 and JC4 could attenuate or slow down retina degeneration in heterozygous $Rho^{P23H/+}$ knock-in mice. These mice feature many hallmarks of human RP [28]. The substitution of Pro23 to His in Rho is the most common mutation responsible for Rho-linked RP [6,29]. We, therefore, injected $Rho^{P23H/+}$ mice with 10 mg/kg of JC3 or JC4 every other day from postnatal (P) day 21 (P21) until P33 followed by the established earlier protocol [15,16], and then used in vivo SD-OCT imaging, histological analyses, and immunostaining to assess the physiological effects of these treatments as shown in (**Fig 8A**). Treatment with these compounds increased the thickness of the ONL in the retina relative to those of the vehicle-treated control mice, which indicates that JC3 and JC4 enhance the survival of photoreceptors (**Fig 8B–8E**). Indeed, the detection of rod cells with an anti-Rho antibody and cone photoreceptors with peanut agglutinin (PNA) confirmed that both JC3 and, to a lesser extent JC4, slowed the progression of photoreceptor cell death in $Rho^{P23H/+}$ mice (**Fig 8D**). The ERG responses of mice treated with JC3 and JC4 exhibited enhanced amplitudes of both dark-adapted and light-adapted responses. However, treatment with JC4 resulted in slightly less

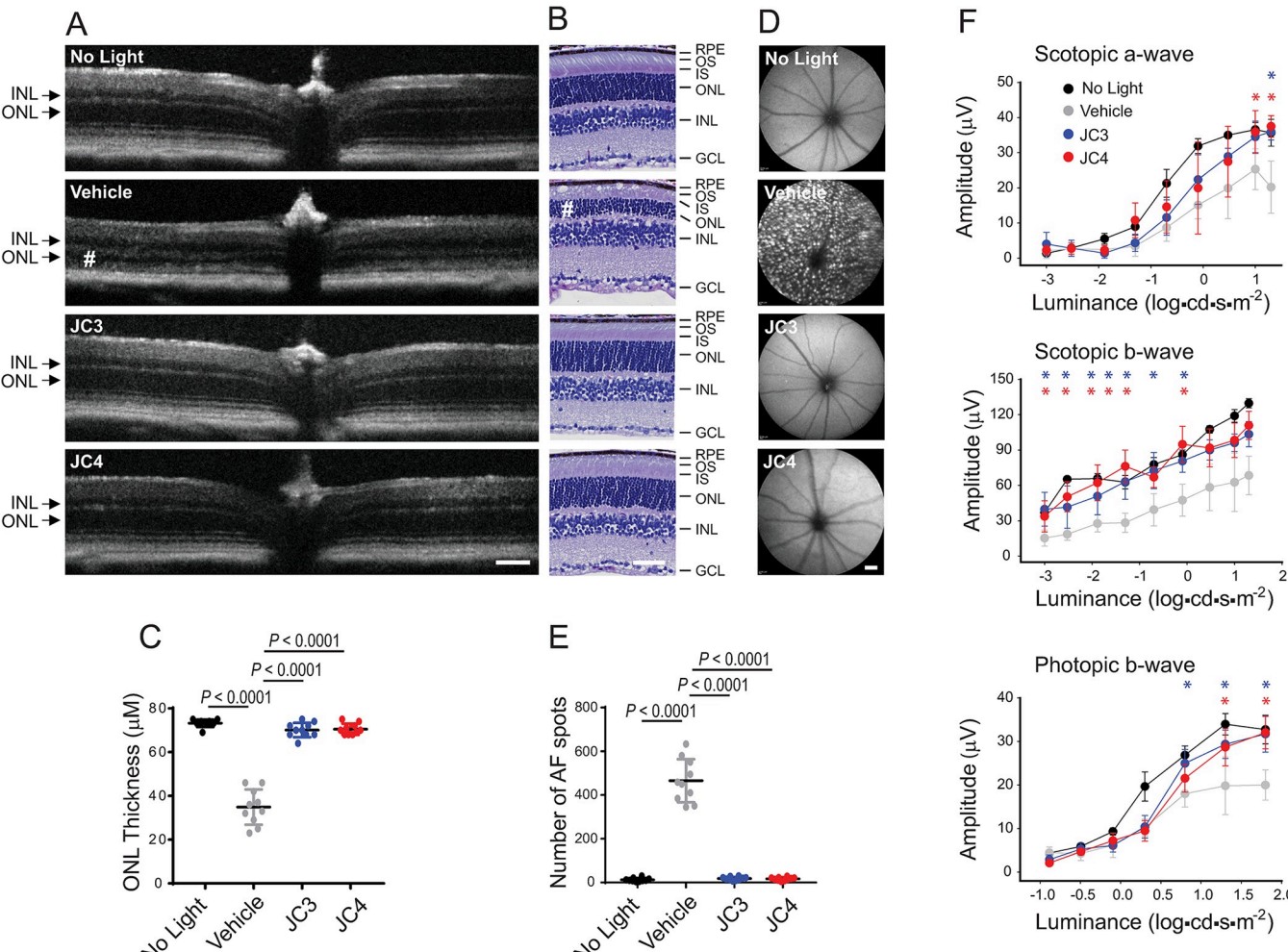

**Fig 7. Protective effect of JC3 and JC4 against retinal degeneration triggered by bright light.** The health of the retina was inspected by SD-OCT and SLO in vivo imaging, and histologically in *Abca4⁻/⁻Rdh8⁻/⁻* mice. Retinal function was examined by ERG. The details of mouse treatment are described in Material and methods. (A) The representative SD-OCT images of the mouse eye after the indicated treatment and exposure to light. SD-OCT, spectral domain-optical coherence tomography; ONL, outer nuclear layer; INL, inner nuclear layer. (#) indicates a disorganized photoreceptor layer in vehicle-treated control mice. Scale bar, 100 μm. (B) Retinal sections stained with HE prepared from eyes collected from mice either unexposed to light or exposed to bright light after the indicated treatment. (#) indicates a disorganized photoreceptor layer in vehicle-treated control mice. Scale bar, 50 μm. (C) Quantification of the ONL thickness measured at 0.5 mm from the ONH ($n$ = 5 mice per treatment group). Error bars indicate SD. The statistically significant changes ($P$) in the ONL thickness observed between dark-adapted and vehicle-treated, exposed to light mice, and between compound-treated and the vehicle-treated group are indicated in the figure. No significant difference in the ONL thickness was observed between mice kept in the dark and those treated with the JC3 and JC4 compounds. (D) The representative SLO images. AF spots were detected only in the retina of DMSO-treated mice injured with bright light. AF spots were not detected in mice kept in the dark or treated with the compounds before illumination. Scale bar, 1 mm. (E) Quantification of AF spots was performed in $n$ = 5 mice per treatment group. Error bars indicate SD. The changes in the number of AF spots after the treatment with JC3 and JC4 compared to vehicle-treated mice were statistically significant ($P$ < 0.001). No significant difference was observed between mice kept in the dark and those exposed to light after treatment with the JC3 and JC4 compounds. (F) The effect of JC3 and JC4 on retinal function in mice injured with bright light ($n$ = 4 for dark adapted mice, $n$ = 7 for vehicle-treated mice, and $n$ = 6 for JC3 and JC4-treated mice). The statistically significant changes ($P$ < 0.05) in the ERG responses obtained upon treatment with JC3 and JC4 are shown with asterisks. Statistical significance was calculated with the two-way ANOVA and post hoc Turkey's tests. The numerical data can be found in S1 Data. AF, autofluorescence; DMSO, dimethylsulfoxide; ERG, electroretinography; HE, hematoxylin and eosin; ONH, optic nerve head; SD, standard deviation; SLO, scanning laser ophthalmoscopy.

pronounced ERG responses (**Fig 8F**). Together, these findings suggest that JC3 and JC4 potentially protect the retina through distinct mechanisms, and the effective dose for JC4 may require additional optimization. Differences in the efficacy of these compounds are likely to arise, in part, from differences in their bioavailability and/or pharmacokinetic profile (**Fig 9**).

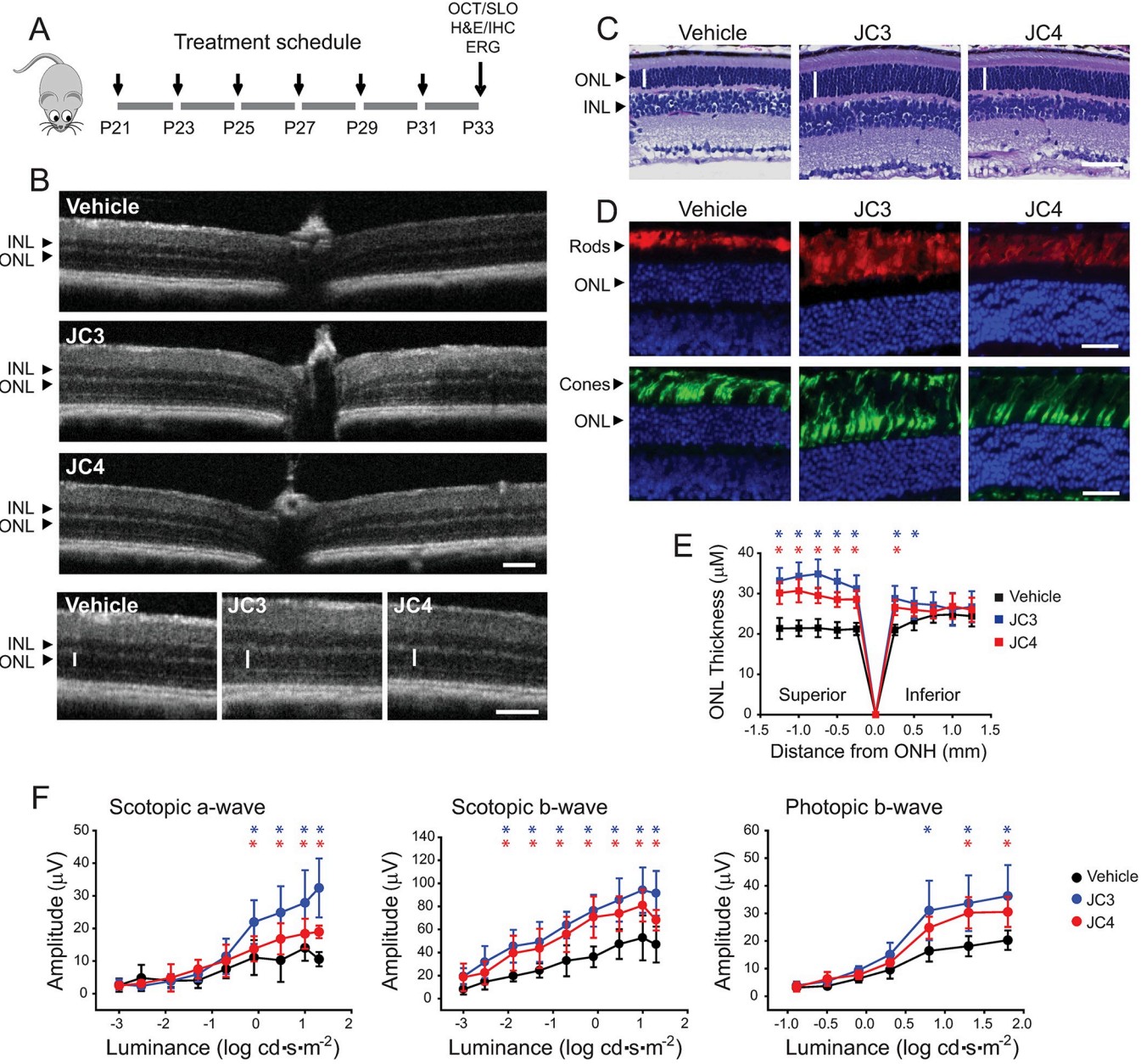

**Fig 8. Protective effect of JC3 and JC4 against retinal degeneration in RP.** The health of the retina was inspected by SD-OCT and SLO in vivo imaging, and histologically in $Rho^{P23H/+}$ mice. Retinal function was examined by the ERG. (A) The treatment strategy with JC3 and JC4 compounds is shown (see details in the Material and methods). (B) The representative OCT images of mouse eyes after the indicated treatment. ONL, outer nuclear layer; INL, inner nuclear layer. Scale bar, 100 μm. (C) Retinal sections stained with HE. Scale bar, 50 μm. (D) The labeling of the retina cryosections with 1D4 anti-Rho antibody and peanut agglutinin to detect rods and cones, respectively. (E) Quantification of the ONL thickness measured at 0.25, 0.5, 0.75, 1.0, and 1.25 mm from the ONH ($n = 7-11$ mice per treatment group). Error bars indicate SD. The statistically significant changes ($P$) in the ONL thickness observed between vehicle-treated and compound-treated mice are indicated in the figure with asterisks. (F) The effect of JC3 and JC4 on retinal function in $Rho^{P23H/+}$ mice ($n = 7-8$ mice per treatment group). The statistically significant changes ($P < 0.05$) in the ERG responses obtained upon treatment with JC3 and JC4 are shown with asterisks. Statistical significance was calculated with the two-way ANOVA and post hoc Turkey's tests. The numerical data can be found in S1 Data. ERG, electroretinography; HE, hematoxylin and eosin; SD, standard deviation; SD-OCT, spectral domain-optical coherence tomography; SLO, scanning laser ophthalmoscopy.

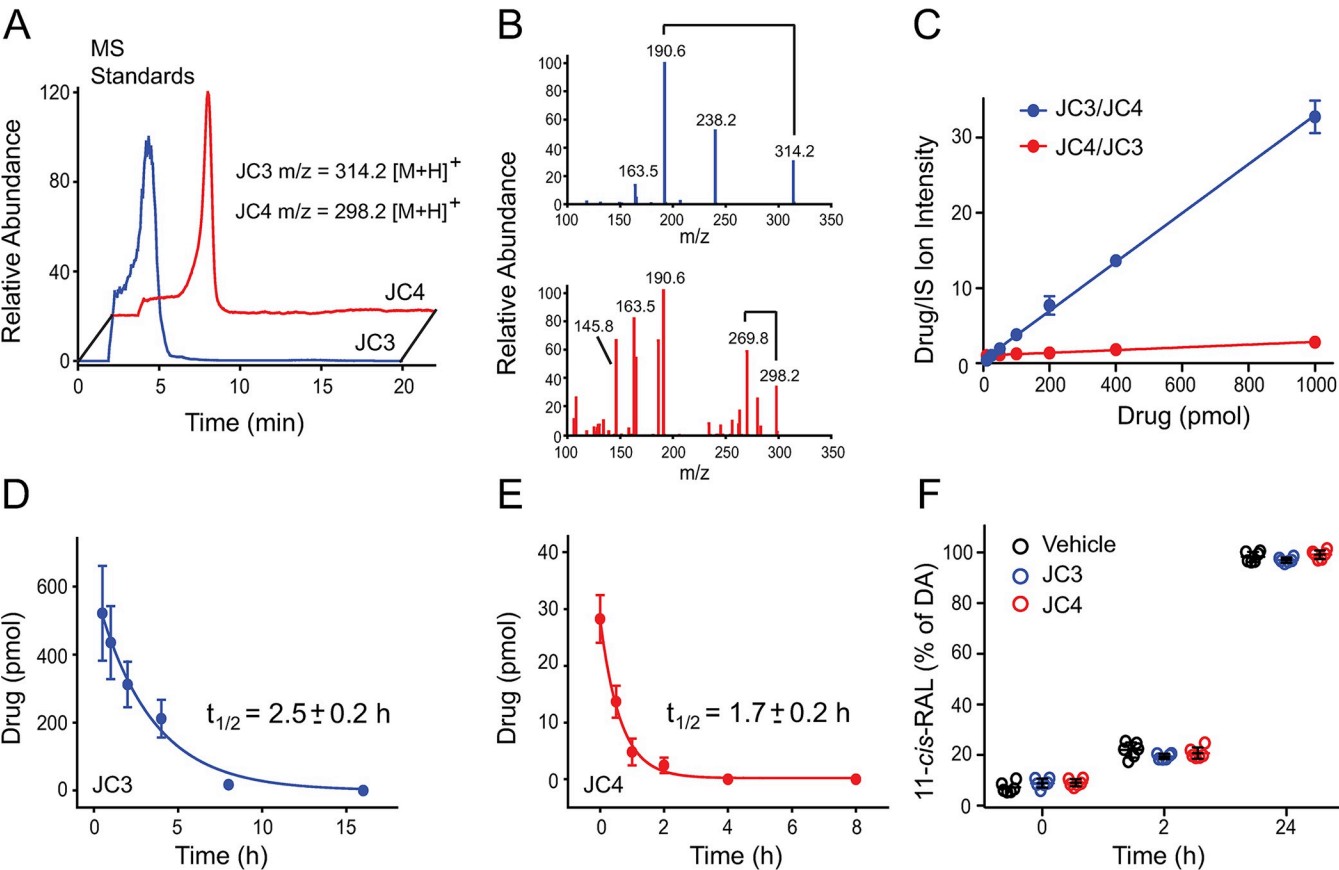

**Fig 9. Detection of JC3 and JC4 in mouse eyes by LC-MS analysis and their effects on the visual cycle.** (A) The HPLC elution profile of JC3 (blue) and JC4 (red). The chromatogram represents ion intensity for m/z = 314.2.2 [M + H]$^+$ corresponding to JC3 and 298.2 [M + H]$^+$ for JC4. (B) The fragmentation patterns of JC3 and JC4 (MS/MS). (C) A standard curve was generated using different concentrations of one compound in the presence of 100 picomoles of the other compound (each compound was used as the internal standard for the other). The correlation for the signal intensity for each combination was determined. The linear regression was obtained from these values, and the concentration of the compounds in the eye was calculated. (D, E) The amount of JC3 and JC4 in the mouse eyes was determined at different time points and their half-lives were calculated. Mice (n = 4) were used per each data. (F) Effect of JC3 and JC4 compounds on the visual cycle. Six-week-old WT C57BL/6J mice were i.p. injected with 100 mg/kg of JC3 (blue), JC4 (red), or vehicle 30 min before exposure to 10,000 lux light for 7 min. Then, these mice were placed in the dark. Eyes were collected from euthanized mice at 0, 2, or 24 h after illumination, followed by the extraction of the retinyl-oximes and their separation by HPLC. The amount of 11-*cis*-retinyl-oxime is presented as a percentage of 11-*cis*-retinyl-oxime detected in vehicle-treated dark-adapted mice and plotted as a function of time after exposure to light illumination. The numerical data can be found in S1 Data. HPLC, high-performance liquid chromatography; WT, wild-type.

Nevertheless, our results indicate that both JC3 and JC4 are good lead compound candidates for the development of next generation more effective pharmacochaperones that stabilize the rod opsin protein, enhance photoreceptor survival, and ultimately slow down the progression of Rho-related RP.

To ensure that advantageous effects of JC3 and JC4 on retina health observed in the heterozygous *Rho*$^{P23H/+}$ mice result from the direct modulation of the rod opsin mutant, we performed treatment of the homozygous *Rho*$^{P23H/P23H}$ mice with these compounds following the earlier established protocol [15]. These mice were administered with JC3 or JC4 every other day starting at P14 and were evaluated at P21. As shown in S2 Fig, this treatment resulted in improved retina morphology evidenced by an increased number of nuclei rows within the ONL layer as compared to the vehicle-treated control mice in both the retina center and periphery (**S2B Fig**). The labeling of rod and cone photoreceptors, while hardly detectable in the vehicle-treated mice, was greatly enhanced in JC3 and JC4-treated mice (**S2A Fig**). The

expression of Rho and M cone opsin was barely detectable in the $Rho^{P23H/P23H}$ mice; however, treatment with JC3 and JC4 resulted in both increased gene and protein expression levels of these receptors, indicating that the JC compounds enhance the stability of the mutant Rho, slowing down degeneration of rod photoreceptors, and consequently cone photoreceptors (**S2C–S2E Fig**). Enhanced survival of photoreceptor cells in $Rho^{P23H/P23H}$ mice treated with JC compounds was confirmed by increased amplitudes of both scotopic and photopic ERG responses as compared to the vehicle-treated mice (**S2F Fig**). This data strongly indicates that the beneficial effects of JC3 and JC4 are related to their pharmacochaperone properties correcting the misfolded Rho mutant and shifting its properties towards WT-like.

## Bioavailability and toxicity of JC3 and JC4

Previous pharmacochaperones for the rod opsin protein have failed to advance to clinical studies due to their toxicity and poor bioavailability [11,12]. To determine whether JC3 and JC4 indeed cross the BRB and persist in the eye in their active forms, we searched for these compounds in biological specimens from treated mice using reverse-phase liquid chromatography coupled with mass spectrometry (LC-MS). C57BL/6J WT mice were treated with a single i.p. injection of each compound prior to harvesting their eyes at various time points. Internal JC3 or JC4 standards were then added to the eye homogenates from mice treated with the opposite compound prior to extraction. An MS signal for the JC3 standard was observed at m/z = 314.2 $[M + H]^+$ along with an MS/MS fragmentation peak at m/z = 238.2 $[M + H]^+$. The MS signal of the JC4 standard was observed at m/z = 298.2 $[M + H]^+$ with a corresponding MS/MS fragmentation product at m/z = 269.8 $[M + H]^+$. Both JC3 and JC4 were detected in the samples extracted from mouse eyes as unmodified compounds (**Fig 9A–9C**). Quantification based on the ion intensities for internal standards revealed 521.1 ± 120.3 pmols of JC3 accumulated in the eye within 30 min of the injection. By comparison, only 28.2 ± 3.2 pmols of JC4 accumulated within the eye during this time. These compounds were cleared from the eye with a calculated half-life of 2.5 h for JC3 and 1.7 h for JC4 (**Fig 9D and 9E**). Together, these results indicate that both JC3 and JC4 can cross the BRB and reach the eye, similarly to other small molecules with retinal protective activity [14,16].

To determine whether the JC compounds are well tolerated in mice, we tested their overall systemic toxicity and their effects on the retinoid (visual) cycle in the eye. The WT mice treated with JC3 or JC4 did not show a decrease in body weight either under acute (at 100 mg/kg) or chronic (at 10 mg/kg) doses administered over the course of 2 weeks (**S3A–S3C Fig**). We also observed no apparent changes in behavior after the chronic administration of JC3 or JC4. A decrease in the exploratory activity observed after the acute administration of JC3 reversed after 2 h. Moreover, these mice treated with JC3 or JC4 for 2 weeks did not exhibit any significant changes in the ERG responses or the ONL thickness in comparison to the vehicle-treated mice, indicating that JC3 and JC4 do not exhibit detrimental effects on the overall retina function and morphology (**S3D–S3F Fig**). Additionally, we did not detect any changes in the levels of regenerated 11-*cis*-retinal at 2 h and 24 h post-illumination among mice treated with JC3 or JC4 compared to the vehicle-treated mice (**Fig 9F**). These findings suggest that these compounds do not inhibit the overall regeneration of the visual chromophore. Together, our results suggest these compounds are well tolerated and accumulate within the eyes of mice without compromising their visual health.

## Structural basis for variant-specific pharmacochaperone effects

Emerging evidence suggests the variant-specific effects of pharmacochaperones arise from differences in their binding energies and the structural context of their binding sites [8,30].

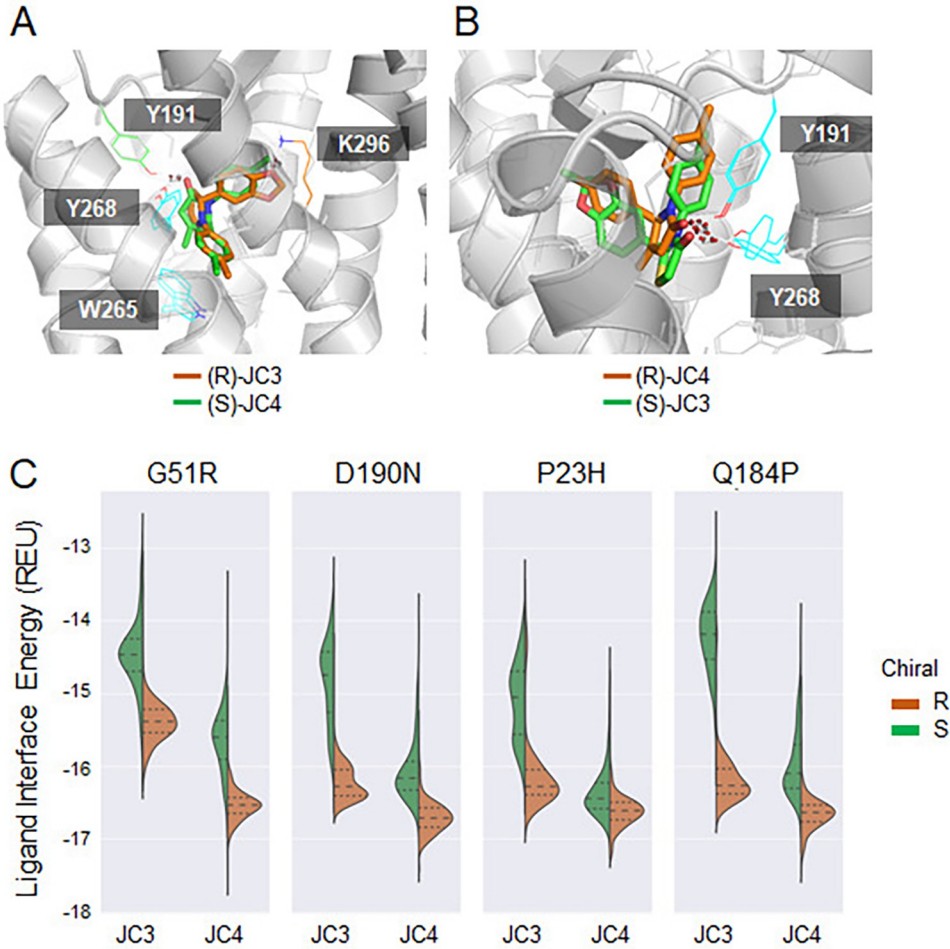

**Fig 10. Structural basis for variant-specific pharmacochaperone effects.** Bound state human WT rod opsin models, highlighted with interacting sites. (A) (R)-JC3 (orange sticks) and (S)-JC4 (green sticks) docked models. (B) (S)-JC3 (green sticks) and (R)-JC4 (orange sticks) docked models. (C) Ligand interface energies in REU of top 10% models for JC3- and JC4-bound variants. WT, wild-type.

Indeed, we observed prominent differences in the response of rod opsin to JC3 and JC4 even though they share a similar pharmacological scaffold, bind with comparable affinity, and compete for the same orthosteric pocket. To evaluate the structural basis for the stabilization generated by these compounds, we employed computational modeling to identify specific interactions that form in the bound state. For this purpose, we used RosettaCM [31] to generate structural models of the human rod opsin apoprotein using the crystallographic structures of the bovine rod opsin apoprotein (93% sequence identity). We then used RosettaLigand [32] to dock each enantiomer of JC3 and JC4 into the orthosteric pocket of the homology model. The lowest energy pose of (R)-JC3 appears similar to that of (S)-JC4 (**Fig 10A**). The methylbenzyl group in (R)-JC3 and the chlorobenzyl group in (S)-JC4 are predicted to have π-stacking with W265 and Y268. Polar interactions are predicted between the benzodioxole in (R)-JC3 and K296 residue, as well as between the carbonyl in (S)-JC4 and Y191 (**Fig 10A**). Likewise, similar low energy poses were also determined for (S)-JC3 and (R)-JC4. The carbonyls in both molecules were predicted to have polar interactions with Y191 and Y268. The methylbenzyl group in (R)-JC4 may also be involved in π-stacking with Y191, while (S)-JC3 is slightly

shifted and excluded from the interaction (**Fig 10B**). These putative differences in the coordination of these 2 compounds may factor into their biological activity.

To determine how these compounds stabilize specific variants, we used these WT models to generate bound-state ensembles of 2 variants that exhibit preferential stabilization by JC3 (G51R and D190N) and 2 that are preferentially stabilized by JC4 (P23H and Q184P). The docked poses and interactions predicted in the WT models were preserved in all variants. These mutations have modest effects on the shape and/or volume of the ligand-binding pocket, except for (S)-JC4 bound G51R where the deeper pocket is closed (**S4 Fig**). The pockets of (R)-JC3 bound models and (S)-JC4 bound models are elongated in a manner that is perpendicular to the plane of the membrane, while those for (S)-JC3 and (R)-JC4 are more compact. For all 4 variants, the R enantiomer of JC3 forms more stabilizing contacts than the S enantiomer (**Fig 10C**). To test this prediction, we assigned the isolated JC3 enantiomers (**S5 Fig**) and then measured their binding affinities (**S6A Fig**). Consistent with modeling results, (R)-JC3 bound to rod opsin with a higher affinity ($K_d = 66.7 \pm 1.3$ nM) relative to (S)-JC3 ($K_d = 333.7 \pm 23$ nM). Moreover, only the R enantiomer was able to enhance the plasma membrane expression of G51R and D190N rod opsins in HEK-293T cells (**S6B Fig**). Thus, we conclude that the R enantiomer of JC3 is the most potent. In contrast, the interface energies formed by the 2 JC4 enantiomers are comparable for each variant except for G51R, where the compounds achieve a distinct pose (**Figs 10C and S4**). Indeed, the isolated JC4 enantiomers appear to bind with similar affinity ($K_d$ values of $91.6 \pm 2.1$ nM and $109.6 \pm 6.1$ nM determined for the 2 unassigned enantiomers). Across all 4 variants, (R)-JC3 and (S)-JC4 have comparable predicted binding energies; however, (R)-JC4 binds with the lowest energy overall. Mutation-specific variations in the energies are generally modest and do not coincide with the selectivity of the variants for one compound over another. Though the similarity of the interaction energies is expected given the comparable binding affinities of these compounds (**Fig 3A**), the modest differences between the bound-state structures of these variants suggest that their divergent pharmacological responses may instead arise from the ability of these compounds to suppress the distinct classes of conformational defects caused by these mutations.

To evaluate the manner in which JC compounds alter the dynamics of pathogenic rod opsin variants, we utilized molecular dynamics (MD) simulations to compare the conformational fluctuations of their bound states over the course of 1,000 ns in an explicit bilayer solvent. The unliganded opsin of the common P23H variant exhibits enhanced flexibility within its N-terminal loop relative to WT, while G51R opsin undergoes dynamic fluctuations in TM1 (**S7A Fig**). MD simulations of these variants bound to (R)-JC3, (S)-JC3, (R)-JC4, and (S)-JC4 showed dampened conformational fluctuations, which suggests that the binding of these compounds partially restores the native structural dynamics of these variants (**S7A Fig**). All 4 compounds remained bound to the orthosteric site over the entire course of the simulations (**S7B Fig**). Based on these simulations, we carried out a PSN analysis, [33] which identified 6 clusters of stabilizing interactions within the native rod opsin structure. The binding of either JC3 or JC4 increased the number of interacting residues within the native clusters or generated additional clusters of interacting residues (**S8 and S9 Figs**). Although we observed minimal differences between the enantiomers in these simulations, generally JC compounds appear to stabilize the native fold by improving internal residue–residue interactions that form in the context of the RP variants.

## Discussion

Proper biogenesis, folding, and stability of rod opsin are required for the correct routing of this receptor to the ROS disc membranes. Mutations in the *RHO* gene cause structural defects

in the receptor protein, which can lead to incorrect folding and defective binding of the native chromophore 11-*cis*-retinal due to weakening internal interactions critical for the intrinsic stability of opsin [15,18]. These changes typically result in the retention and degradation of the mutant receptor in the endoplasmic reticulum (ER). In P23H Rho mice, the mutant receptor escapes ER quality control and is transported to the ROS, but ultimately compromises the integrity of its membrane discs in a manner that causes autosomal dominant RP (adRP) [28,34]. This pathology could potentially be rescued using pharmacological chaperones, which may include its native retinal chromophore or its analogs that restore opsin stability [11]. However, stabilization and folding correction of misfolded opsin variants with retinoids can only occur under dark conditions due to the photosensitivity of the Schiff base bond formed between the retinal and the opsin's Lys296. The utility of retinoids is also undermined by the fact that these compounds and their photoproducts are toxic, which limits their therapeutic potential. For these reasons, the use of commercially available and commonly used in in vitro assays 9-*cis*-retinal is not desirable, and thus the development of novel non-retinoid folding correctors that are insensitive to light provides a clear advantage in ongoing efforts to develop pharmacochaperones for the rod opsin protein. In fact, small molecule correctors of protein folding have been proven effective in other protein misfolding diseases [35–37].

We recently described a series of flavonoids and other unrelated small molecules that are capable of stabilizing the opsin protein and acting as pharmacochaperones under certain conditions [14–16]. Nevertheless, these compounds exhibit limited bioavailability and appear to only partially correct the misfolding of certain RP variants [16]. In this study, we describe 2 new pharmacochaperones, JC3 and JC4, which bind reversibly to ligand-free opsin and modulate opsin properties without regulation by light. These molecules increased the stability of opsin and substantially improved the maturation and expression of the heterologously expressed P23H rod opsin mutant. Thermal stability assay indicated that JC3 and JC4 did not provide additional stabilization in the presence of 9-*cis*-retinal, which suggests they likely bind to the same orthosteric binding pocket. Indeed, the binding of JC compounds partially antagonized the coupling of the 9-*cis*-retinal isochromophore, slowing down the rate of pigment regeneration without inhibition of isoRho formation. This decelerated regeneration of the visual receptor could potentially be beneficial under pathological conditions in which the clearance of cytotoxic all-*trans*-retinal photoproducts is impaired, which is known to occur in Stargardt disease and age-related macular degeneration (AMD) [38]. These excessive concentrations of all-*trans*-retinal induce a cellular stress response within the retina, which has detrimental consequences for photoreceptors. Slower formation of the functional Rho pigment could also attenuate the production and release of all-*trans*-retinal upon illumination. Indeed, we showed that JC3 and JC4 prevented photoreceptor death that occurs in response to bright light injury in an $Abca4^{-/-}Rdh8^{-/-}$ mouse model of Stargardt disease, likely due to their stabilization of opsin and their ability to antagonize opsin constitutive activation under these conditions, which is known to accelerate degenerative processes in the retina [24,25].

In addition to their photosensitivity and limited bioavailability, previously described rod opsin pharmacochaperones generally appear to only partially rescue a subset of known retinopathy variants [8,16]. Using deep mutational scanning, we showed that in total, JC3 and JC4 measurably enhanced the plasma membrane expression of 36 of 123 heterologously expressed RP-related rod opsin mutants. Notably, 22 of these mutants appear to be sensitive to both compounds. Interestingly, treatment with JC3 enhanced the plasma membrane expression of 8 distinct mutants, while JC4 appears to be uniquely effective towards 4 other distinct rod opsin mutants. Although JC3 and JC4 restore the expression of fewer mutants than 9-*cis*-retinal under these conditions [8], we suspect this limitation could potentially be overcome by modifications that enhance the affinity of these compounds for the rod opsin protein [3]. Regardless

of whether these compounds could be modified to correct the expression of a broader array of mutants, we noted that several of these mutants exhibit a response to JC compounds that is comparable in magnitude to the effect of 9-*cis*-retinal or even greater. Together, the observed differences in the response of this panel of mutants to retinoids, chromenones, and JC compounds suggest that, regardless of the specific details of the chemical scaffold, this class of molecules may eventually need to be targeted to specific patient genotypes in clinical trials. The development of a diverse portfolio of rod opsin pharmacochaperones may therefore be beneficial for the development of precision therapeutics, as has proven successful for CF [39].

While the in vitro studies described herein provide insights into the mechanistic effects of these compounds, the phenotypic effects of JC3 and JC4 in $Abca4^{-/-}Rdh8^{-/-}$ and $Rho^{P23H/+}$ mice provide compelling evidence of the therapeutic potential of these compounds. Importantly, treatment with either compound improved the overall retina health and function in these mice by prolonging the survival of their photoreceptors. The beneficial effect of JC3 and JC4 on the photoreceptor survival in homozygous $Rho^{P23H/P23H}$ mice, showing similar outcomes as reported previously for treatment with flavonoids, [15] confirms the pharmacochaperone potential of these compounds for the RP-linked rod opsin mutant. We noted that, although JC4 appeared to be most effective in vitro, JC3 proved more effective in vivo, especially in $Rho^{P23H/+}$ mice, likely due to its longer half-life in the eye. Furthermore, neither of these compounds caused significant toxicity in mice nor had adverse effects on retina morphology or function in the WT mice. Based on these considerations, the 2 compounds described herein offer considerable translational potential for the development of precision therapeutics for RP and other visual retinopathies.

Nevertheless, 2 primary limitations of the JC compounds can be highlighted, relatively short half-life and limited solubility. Thus, additional structure-activity relationship (SAR) studies would be beneficial to further refine and optimize the pharmacokinetic (PK) and pharmacodynamic (PD) profiles of these compounds, potentially leading to development of improved candidate molecules. Expanding the chemical diversity could also enable the development of derivative molecules with better solubility and comparable or improved potency. Thus, additional investigations are needed to determine whether the efficacy of one or both of these compounds can be enhanced through modification of the structure and/ or dosing regimen in vivo.

Moreover, further studies examining the effects of JC compounds on cellular signaling pathways are essential to elucidate their precise mechanisms of action. It is likely that these compounds, by directly interacting with the mutant receptor, enhance its proper folding and stability, resulting in reduced ER stress and attenuating the unfolded protein response (UPR) pathway activation. Diminishing these stress response processes could in turn lessen apoptotic signaling, enhancing photoreceptor survival.

## Materials and methods

### Chemicals and reagents

The 4969-Diamidino-2-phenyl-indole (DAPI) for nuclear staining was purchased from Life Technologies (Grand Island, New York, United States of America). Dimethylsulfoxide (DMSO) was obtained from Sigma (St. Louis, Missouri, USA). EDTA-free protease inhibitor cocktail tablets were purchased from Roche (Basel, Switzerland), and 9-*cis*-retinal was purchased from Sigma. BODIPY FL L-cystine (BFC) was obtained from Thermo Fisher (Waltham, Massachusetts, USA). Polyvinylidene difluoride (PVDF) membrane was obtained from Millipore (Burlington, Massachusetts). JC3 was purchased from ChemBridge (San Diego). JC4 was obtained from MolPort (Latvia). Larger amounts of JC3 and JC4 were custom-synthesized

by Wuxi AppTech (China). JC3 and JC4 unassigned enantiomers also were purchased from Wuxi AppTech (China).

## List of antibodies

Anti-GAPDH, mouse monoclonal, Abclonal, NO: AC002, dilution 1:10,000

Anti-mouse IgG, HRP conjugate, goat, Promega, No: W4021, dilution 1:10,000

Anti-rabbit IgG, HRP conjugated, goat, Promega, No: W4011, dilution 1:10,000

Anti-mouse IgG Alexa Fluor 555-conjugated, goat, Thermo Fisher, No: A28180, dilution 1:400

Anti-rabbit IgG Alexa Fluor 555-conjugated, goat, Thermo Fisher, No: A27039, dilution 1:400

Anti-rabbit IgG Alexa Fluor 594-conjugated, goat, Thermo Fisher, No: A11005, dilution 1:200

Dylight 550-labeled anti-HA, mouse, Thermo Fisher, No: 2–2.2.14, dilution 1:100.

## Virtual screening

The computational screening of purchasable compounds available in the *Zinc* database (http://zinc.docking.org) was performed against the chromophore-binding site of rod opsin. The monomeric unit of the crystal structure of bovine rod opsin (PDB ID: 3CAP) was used for the molecular docking experiment. Water and other co-crystallized molecules were removed from the coordinate set, the hydrogen atoms were added, and partial charges were assigned to all atoms. The protein then was subjected to restrained molecular mechanics refinement with NAMD 2.12 software using the CHARMM22 force field. The coordinates for the ligand-binding pocket located in the orthosteric site were selected as reported in [12]. Compounds then were docked within the orthosteric-binding site using VINA/Vega 3.1.0.21 software, and the binding free energies were calculated in kcal/mol. The docking results were visualized using Biovia Discovery Studio Visualizer 17.2.0 software to assess binding poses within the orthosteric site and identify interactions within the binding pocket. The docking results obtained were then rescored by docking each compound 10 times. Binding poses and interactions (location into the binding site and number and type of interactions formed) were compared using Biovia Discovery Studio Visualizer 17.2.0 software.

## Pharmacokinetic drug properties

An analysis of physicochemical descriptors, parameters related to ADME, and drug-likeness of the compounds was carried out using SWISSADME tools (http://www.swissadme.ch) [19]. The results are presented in **Fig 1**.

## Preparation of opsin membranes

Bovine retinas were used to isolate the rod outer segment (ROS) membranes and opsin membranes were prepared as previously described [16,40]. Membranes were washed 4 times with a hypotonic buffer containing 5 mM HEPES (pH 7.5) and 1 mM EDTA then pelleted by centrifugation at 25,000×g for 25 min. The final membrane pellet was suspended in 10 mM sodium phosphate (pH 7.0) and 50 mM hydroxylamine at Rho concentrations of ~2 mg/ml. Membranes were then exposed to light with a 150 Watt bulb for 30 min at 0˚C. Then, membranes were pelleted at 16,000×g for 10 min in the bench-top centrifuge. The membrane pellet was washed twice with 10 mM sodium phosphate (pH 7.0) and 2% BSA followed by 4 washes with

10 mM sodium phosphate (pH 7.0) and 2 washes with 20 mM BTP (pH 7.5) containing 100 mM NaCl. After each wash, the membranes were centrifuged at 16,000×g for 10 min at 4˚C.

The concentration of opsin and Rho was determined after ROS membrane solubilization with 20 mM dodecyl-β-D-maltopyranoside (DDM) and pelleting insoluble material at 16,000×g for 15 min at 4˚C using a UV-visible spectrophotometer (Cary 60, Varian, Palo Alto, California, USA). The absorption coefficients $\varepsilon_{280nm}$ = 81,200 $M^{-1}cm^{-1}$ for opsin, $\varepsilon_{500nm}$ = 40,600 $M^{-1}cm^{-1}$ for Rho, and $\varepsilon_{485nm}$ = 43,600 $M^{-1}cm^{-1}$ for isoRho were used to calculate the concentration [22,41,42].

### Detection of compound binding to opsin

UV-visible spectroscopy–Washed ROS membranes suspended in the buffer consisting of 20 mM BTP (pH 7.5) and 100 mM NaCl at a final opsin concentration of 5 μM were incubated with 10 μM of each compound for 30 min at room temperature prior to incubation with 5 μM 9-*cis*-retinal for 15 min. For comparison, membranes were incubated only with 9-*cis*-retinal. Next, 20 mM DDM detergent was added to the membrane suspension. Following a 5-min incubation at room temperature, the samples were centrifuged at 16,000×g for 5 min at 4˚C. The UV-visible spectra were measured.

Fluorescence spectroscopy–The intrinsic tryptophan fluorescence quenching was used to probe the binding of the compounds within the orthosteric binding pocket of rod opsin. Opsin membranes at 4.2 nM were solubilized in the buffer containing 20 mM BTP (pH 7.5), 100 mM NaCl, and 1 mM DDM. Tryptophan fluorescence was monitored before and after incubation with compounds at 0.1, 0.2, 0.3, 0.4, 0.5, 1.0, and 1.5 μM concentrations. The ligands were incubated with opsin for 2 min before the measurements. An FL 6500 Fluorescence Spectrometer was used to record the emission spectra at 20˚C between 300 and 420 nm after excitation at 295 nm. The excitation and emission slit bands were set at 5 and 10 nm, respectively. The changes in the intrinsic tryptophan fluorescence at 330 nm ($\Delta F/F_0$, where $\Delta F$ is the difference between the initial Trp fluorescence ($F_0$) and fluorescence recorded upon addition of the compound) were plotted as a function of the ligand concentration. The ligand-binding curves were fitted and binding affinities ($K_d$) were calculated using GraphPad Prism 7.02 software. All measurements were performed in triplicate. The experimental data were corrected for the samples' background and self-absorption at excitation and emission wavelengths (inner filter effect correction).

### Pigment regeneration assay

ROS opsin membranes at 5 μM concentration were incubated with the compounds at 1, 10, and 100 μM concentrations for 30 min at room temperature prior to membrane solubilization with 20 mM DDM for 5 min at room temperature. Solubilized opsin was cleared by centrifugation at 16,000×g for 5 min at 4˚C, and 5 μM 9-*cis*-retinal was then added to the sample and UV-visible spectra were measured every 2 min for 60 min at 20˚C. Each condition was repeated 3 times. The absorbance at 485 nm was plotted as a function of time and a time course of pigment regeneration was fitted to a second-order exponential decay to calculate the rates and the apparent half-lives of isorhodopsin (isoRho) regeneration.

### Thermal stability

To compare their effect on pigment stability, isoRho's regenerated with 9-*cis*-retinal following treatment with JC3 and JC4 were incubated at 55˚C in the dark. Their UV-visible spectra were then recorded every 2 min for 1 h. The absorbance at 485 nm was plotted as a function of time,

and the half times ($t_{1/2}$) of the chromophore release were calculated by fitting the observed decay with a single exponential rate equation. All samples were measured in triplicate.

Thermal shift assays were used to examine the effect of JC3 and JC4 on the stability of unliganded opsin. The opsin membranes were suspended in 20 mM BTP, pH 7.5, and 100 mM NaCl at a concentration of 0.01 mg/ml and 20 μl were pipetted into a 96-well plate (Applied Biosystem). Indicated compounds were added to final concentrations of 0.1, 1, 10, 100 nM, 1, and 10 μM and incubated for 1 h at 4°C; 5 μl of the BODIPY FL L-cystine (BFC) probe (Thermo Fisher) was then added to each well. Opsin membranes without treatment were included as a control. The plate was sealed with a ClearSeal film (HR4-521, Hampton Research) and incubated for 10 min on ice before the measurement. The changes in the sample fluorescence were measured with a StepOnePlus Real-Time PCR System (Applied Biosystems) and analyzed using the StepOne software version 2.3. The fluorescence in the SYBR, FAM, and ROX channels was recorded for each sample. Cycles included an initial cooling step at 4°C for 1 min then warmed by 1°C and held at that temperature for 1 min until the sample reached 99.9°C. Multicomponent data were then analyzed using GraphPad Prism 7.02 software. The melting temperatures ($T_m$) of bovine Rho and opsin within the membranes were 71.9°C and 55.4°C, respectively [43]. Each condition was repeated in triplicate.

### $G_t$ activation assay

The $G_t$ protein was extracted and purified from ROS membranes isolated from a hundred dark-adapted bovine retinas [14,44]. The effect of tested compounds on Rho function was tested by measuring the changes in the intrinsic tryptophan fluorescence of $G_{t\alpha}$. The opsin membranes at 50 nM concentration suspended in 20 mM BTP (pH 7.0) containing 120 mM NaCl and 1 mM $MgCl_2$ were incubated with a 10 μM compound for 30 min at room temperature prior to regeneration of the pigment with 5 μM 9-*cis*-retinal for 10 min at room temperature. Next, $G_t$ was added to 500 nM concentration and the sample was illuminated for 1 min with a Fiber-Light illuminator (Dolan Jenner Industries Inc., Boxborough, Massachusetts, USA) through a 480 to 520 nm band-pass wavelength filter (Chroma Technology Corporation, Bellows Falls, Vermont, USA). Next, 10 μM GTPγS was added and the measurement was performed for 1,200 s with an FL 6500 Fluorescence Spectrometer. Excitation and emission wavelengths were set at 300 nm and 345 nm, respectively [14,45]. $G_t$ activation rates were determined for the first 500 s.

### Cell culture

The NIH-3T3, HEK-293, and ARPE19 cells were cultured in DMEM with 10% FBS (Hyclone, Logan, Utah, USA), and 1 unit/ml penicillin with 1 μg/ml streptomycin (Life Technologies) at 37°C under 5% $CO_2$ according to the instructions from the ATCC Animal Cell Culture Guide. The 661W cells, murine photoreceptor-derived cells, were provided by Dr. Muayyad Al-Ubaidi, University of Houston, and cultured in DMEM with 10% FBS (Hyclone), and 1 unit/ml penicillin with 1 μg/ml streptomycin (Life Technologies) at 37°C under 5% $CO_2$ according to the received instructions.

### Cytotoxicity assay

Cells were plated in the 96-well plate at a density of $3 \times 10^5$ cells/well and 24 h later were treated with different concentrations of the identified compounds for an additional 24 h. The 3-(4,5-Dimethyl-2-thiazolyl)-2,5-diphenyl-2H-tetrazolium bromide (MTT) cell proliferation assay (Sigma) was used to assess the effect of these compounds on cell viability. Non-treated cells were used as a control. Cytotoxicity was determined by calculating the percentage of dead

cells in each experimental condition. All experimental conditions were performed in triplicate and the experiments were repeated 3 times.

## Detection of rod opsin in the cell membrane and signaling

The NIH-3T3 cells stably expressing rod opsin and GFP were plated in the 96-well plate at a density of $3 \times 10^5$ cells/well. Compounds were added to a final concentration of 10 μM for 6 h after plating and the cells were then incubated for 16 h. Cells treated with 5 μM 9-*cis*-retinal were included as a positive control. The next day, cells were fixed with 4% paraformaldehyde for 20 min at room temperature and then washed twice with phosphate buffer saline (PBS). Cells were then incubated with 10% normal goat serum in PBS for 1 h at 37˚C prior to incubation with B6-30 anti-Rho antibody recognizing the receptor N-terminus for 2 h at 37˚C. The cells were then washed 3 times with PBS prior to incubation with an anti-mouse antibody conjugated with Alexa Fluor 594 (Thermo Fisher Scientific) at 1:200 dilution for 1 h at room temperature. Cells were then washed with PBS 3 times and the nuclei were stained with DAPI following the manufacturer's protocol. Finally, the plate was sealed with a ClearSeal film (HR4-521, Hampton Research) and used for cell imaging. Cells were imaged using an Operetta High Content Imager (Perkin Elmer Life Sciences). DAPI fluorescence was used to define nuclei and count cells. Bright-filed images and GFP fluorescence were used to define cell bodies. The plasma membrane fluorescence was defined within ±5% of the cell border. In addition, photoreceptor-derived 661W cells stably expressing P23H rod opsin were used.

## cAMP detection assay

The NIH-3T3 cells stably expressing either WT or P23H rod opsin were plated in two 96-well plates at a density of 50,000 cells per well in 85 μl of DMEM medium containing 10% FBS and antibiotics. Cells were treated with compounds at different concentrations after 6 h and incubated for 16 h. The next day, 9-*cis*-retinal was added for 2 h in the dark to regenerate isoRho. Then, one plate was kept in the dark, while the second plate was exposed to bright (150 Watt) light for 15 min from a 10 cm distance. The cAMP-Glo assay (Promega) was used to detect the levels of accumulated cAMP following the manufacturer's protocol. The luminescence signal was recorded with a FlexStation 3 plate reader (Molecular Devices). A standard curve was prepared using cAMP provided by the kit and the percentage of cAMP was calculated for each condition. The values were expressed as a percentage, assuming the cAMP level detected in the non-treated cells as 100%.

## Mutational scanning

We compared the effects of JC3 and JC4 compounds on the plasma membrane expression of 123 retinopathy variants in parallel using deep mutational scanning [8]. Briefly, a mixed recombinant stable HEK-293T cell line in which each individual cell inducibly expresses one of 123 known retinopathy variants was generated as previously described [8,46,47]. An N-terminal hemagglutinin epitope tag was then used to mark the expressed Rho variants at the plasma membrane by surface immunostaining using a Dylight 550-labeled anti-HA antibody (Thermo Fisher). Cells were then separated based on their relative surface immunostaining using fluorescence-activated cell sorting (FACS). Cellular isolates were then expanded prior to the extraction of the recombined genomic DNA from each fraction. Illumina sequencing was then used to characterize each cellular isolate by quantifying the relative abundance of a series of unique molecular identifiers associated with each variant. The relative surface immunostaining of each variant was then inferred from the sequencing data in the presence and absence of JC3, JC4, or 9-*cis*-retinal. The analysis was repeated for each condition.

## Structural modeling and ligand docking

Comparative modeling of human rod opsin receptor–To model human rod opsin in its ligand-free state, the crystal structure of bovine rod opsin (PDB ID: 3CAP) was used as a template (93% sequence identity). Comparative modeling was carried out with RosettaCM [32] following a published protocol for G-protein coupled receptors [48–53]. Sequence alignment was done with ClustalOmega using default settings [49]. Membrane topology files are generated with TOPCONS [50] and only OCTOPUS [51] results were kept and converted to Rosetta readable span files using Rosetta built-in octopus2span.pl script. Coordinates of human rod opsin were then generated from aligned templates with Rosetta partial_thread application. The threaded models were hybridized and relaxed through Rosetta XML scripts [52]. Since the template sequences are highly similar to human opsin sequence, hybridization created 2,000 models and the relaxed models were ranked by total_score. The best-scoring model was selected for later docking studies.

Ligand docking into wild-type homology models–The ligand docking protocol applied here was modified from RosettaLigand standard docking protocol [32]. Ligand conformers were generated with Biochemical Library (BCL) using default settings of BCL::Conf [53]. The retinal orthosteric-binding site was chosen as the pocket for ligand docking. Low-resolution docking phase allowed to sample ligand-binding modes within 5.0 Å from the center of the pocket. High-resolution phase performed 6 cycles of flexible sampling of sidechain rotamer and ligand conformer. The structure was then relaxed with the ligand and protein non-neighbor (10.0 Å or more away from the ligand) atoms fixed after low-resolution and high-resolution phases. Before the final minimization, another round of high-resolution docking phase was carried out to refine the binding mode. The RosettaLigand energy function was used. This protocol was repeated to generate 10,000 models for each enantiomer of JC3 and JC4. The models were sorted by interface delta score and the top 5% models were clustered into 3 groups based on root mean square deviation (RMSD). The model for further analysis were manually selected from top models of each cluster after visual inspection.

Ligand docking into mutation variants–The top-scoring docked wild-type models were mutated with Rosetta MutateResidue mover. Another phase of high-resolution docking was then applied to the mutated complex, followed by relaxation with fixed ligand atoms and final minimization. This protocol was repeated to generate 5,000 models for each variant. The models were first sorted by total score and the top 10% models were then sorted by interface delta score. The top models were clustered into 3 groups based on RMSD and top models of each cluster were visually inspected.

## MD simulations and PSN analysis

Models of selected human opsin variants and compound complexes were used for MD simulations (Desmond Molecular Dynamics System, D. E. Shaw Research, New York, NY, 2020 and Maestro-Desmond Interoperability Tools, Schrödinger, New York, NY, 2020) [15,33,54–57]. The protein-compound complexes were inserted in a membrane of POPC (1-palmitoyl-2-oleoyl-sn-glycero-3-phosphocholine) [54] at 300K, while the transmembrane region was obtained following the data of UniProt database (https://www.uniprot.org/). The complexes created were immersed in a box filled with water molecules using the simple point charge (SPC) scheme. The dimension of the solvent buffer was set to 10 Å$^3$. Counter ions (2 Na$^+$) were added to neutralize charges and additional Cl$^-$ and Na$^+$ ions were added to obtain a final NaCl concentration of 150 mM. Energy minimization was carried out by 2,000 steps using the steepest descent method with a threshold of 1.0 kcal/mol/Å. Periodic boundary conditions were used and a cutoff of 9 Å was established for van der Waals interactions and the particle

mesh Ewald (PME) method with a tolerance of $10^{-9}$ was used in the electrostatic part. The NPT simulations were realized at 300 K with the Nosé–Hoover algorithm [56] and the pressure was maintained at 1 bar with the Martyna–Tobias–Klein barostat [55]. The OPLS3e force field was used in all runs [57]. The simulation length was 1,000 ns. The final states of the proteins were analyzed by the PSN using the software webPSN [15,33]. This tool allows inferring the hubs, links, and communities between the residues of the protein.

## Stereochemistry of JC compounds

Stable stereoisomers of JC3 were obtained with high purity, JC3-1 (98.5%) and JC3-2 (99.5%). However, only one JC4 enantiomer, JC4-1 (99.5%) was stable, while JC4-2 rapidly converted back to a racemic mixture (62.5% compound enrichment). To assign the stereochemistry of the JC3 isolates, we collected circular dichroism spectra for each compound at 1 μM in acetonitrile using a Jasco J-1500 CD Spectrometer (Jasco, Oklahoma City, Oklahoma, USA). To match these spectra to specific isomers, we then used the Gaussian 16 suite [58] to model the electronic structure of each isomer and simulate their ECD spectra. The geometric optimization and frequency calculations were carried out with the 6-311G(d,p) Pople basis set and wB97X-D functional. The ECD calculations were obtained by performing the TD-DFT calculation method on the geometrically optimized structures at the same level of theory in acetonitrile using the SMD solvation model [59]. The ECD calculations were obtained by performing the TD-DFT calculation method on the geometry-optimized structure at the same level of theory and spectra for individual conformers were Boltzmann weighted to obtain the final mixture ECD spectrum.

## Mouse models

Six-week-old $Abca4^{-/-}Rdh8^{-/-}$ mice (Research Resource Identifier, **RRID**: IMSR_JAX:030503, The Jackson Laboratory, Bar Harbor, Maine, USA) were used to test the protective effects of JC3 and JC4 compounds against acute light-induced retinal degeneration. $Abca4^{-/-}Rdh8^{-/-}$ mice were genotyped to confirm that they do not carry the $Rd8$ mutation. These mice carry the Leu variation at amino acid 450 of retinal pigment epithelium 65 kDa protein ($RPE65$). Substitution of Leu to Met decreases sensitivity to light-induced photoreceptor degeneration [60,61]. Heterozygous $Rho^{P23H/+}$ knock-in mice were used to evaluate the effectiveness of JC3 and JC4 in RP. To obtain heterozygous $Rho^{P23H/+}$ mice, $Rho^{P23H/P23H}$ (**RRID**: IMSR_JAX:017628, The Jackson Laboratory, Bar Harbor, Maine, USA) were crossed with WT C57BL/6J mice (**RRID**: IMSR_JAX:000664, The Jackson Laboratory, Bar Harbor, Maine, USA). Compounds dissolved in 50% DMSO in PBS were administered to mice by intraperitoneal (i.p.) injection. Both male and female mice were used in all experiments. All mice were housed in the Animal Resource Center at the School of Medicine, Case Western Reserve University (CWRU) and maintained in a 12-h light/dark cycle. All animal procedures and experimental protocols were approved by the Institutional Animal Care and Use Committee (IACUC) at CWRU and conformed to recommendations of both the American Veterinary Medical Association Panel on Euthanasia and the Association for Research in Vision and Ophthalmology as well as the National Eye Institute Animal Care and Use Committee (NEI-ASP 682). The protocol number is 2015–0124. Efforts were taken to minimize animal suffering.

## Animal treatment

The $Abca4^{-/-}Rdh8^{-/-}$ mice were dark-adapted 24 h before the treatment. The compounds at a concentration of 100 mg/kg body weight (b.w.) or DMSO vehicle were delivered to mice 30 min before the exposure to bright light. Mice pupils were dilated with 1% tropicamide and the

retinal degeneration was initiated by illumination of mice with 10,000 lux light (150-W bulb, Hampton Bay, Home Depot, Atlanta, Georgia, USA) for 30 min. Retinal structures were visualized and analyzed in vivo by ultra-resolution SD-OCT (Bioptigen, Morrisvill, North Carolina, USA) and SLO (Heidelberg Engineering, Franklin, Massachusetts, USA). Retinal function was examined with ERG. Analyses were performed 7 to 10 days after light exposure. Mice were euthanized by cervical dislocation under deep anesthesia with a cocktail containing ketamine (20 mg/ml) and xylazine (1.75 mg/ml) at a dose of 4 μl/g b.w. Eyes were collected for preparing paraffin sections, which were used for staining with hematoxylin and eosin (HE) or cryosections.

$Rho^{P23H/+}$ mice [28] were used to determine the effects of the JC3 and JC4 on the progression of retinal degeneration in RP as described in [15,16]. Compounds at 10 mg/kg b.w. or vehicle were administered i.p. to mice starting at postnatal (P) day 21 (P21). A total of 6 injections were performed every other day at 3:00 PM. Retinal morphology was visualized with the SD-OCT and retinal function was examined with ERG. Before each procedure, mice were anesthetized with a cocktail containing ketamine (20 mg/ml) and xylazine (1.75 mg/ml) at a dose of 4 μl/g b.w. For histological and immunohistochemical examinations, eyes were collected from mice euthanized by cervical dislocation under deep anesthesia.

In addition, $Rho^{P23H/P23H}$ mice were treated with JC compounds at 10 mg/kg b.w. or vehicle every other day (3 injections) between P14 and P21.

To assess the potential toxicity of JC3 and JC4 WT C57BL/6J mice were treated either with a single dose of these compounds (100 mg/kg) at P33 or 6 doses (10 mg/kg) administered every other day between P21 and P33. The body weight of these mice was monitored every other day for 2 weeks. Additionally, the ERG and SD-OCT measurements were performed at the end of prolonged compound administration.

## In vivo imaging of mouse retina

The protective effects of the identified compounds against retinal damage induced by acute light in $Abca4^{-/-}Rdh8^{-/-}$ and inherited mutation in $Rho^{P23H/+}$ mice were evaluated by in vivo imaging with the scanning laser ophthalmoscopy (SD-OCT). Before imaging, mice pupils were dilated with 1% tropicamide and mice were anesthetized by i.p. injection of a cocktail containing ketamine (20 mg/ml) and xylazine (1.75 mg/ml) at a dose of 4 μl/g b.w. The a-scan/b-scan ratio was set at 1,200 lines. The SD-OCT retinal images were obtained by scanning at 0 and 90 degrees in the b-mode. Five image frames were captured and averaged. The changes in the retinas of mice exposed to bright light and control mice were determined by measuring the ONL thickness at 0.5 mm from the optic nerve head (ONH). Values of the ONL thickness were plotted using means and standard deviation. Each experimental group contained at least $n = 6$ mice.

The in vivo whole-fundus imaging of mouse retinas in $Abca4^{-/-}Rdh8^{-/-}$ mice was performed with the SLO. After the SD-OCT imaging, mice immediately were subjected to the SLO imaging. Images were collected in the auto-fluorescence mode. The number of autofluorescence spots (AF) detected was counted, and the data were analyzed to determine the statistical significance. Each experimental group contained at least $n = 5$ mice.

## Retina histology

After the in vivo imaging, mouse eyes ($n = 6$ per group) were collected from euthanized mice and fixed in 0.5% glutaraldehyde in 2% paraformaldehyde in PBS for 24 h at room temperature on a rocking platform. Then, the fixation solution was changed to 1% PFA for 48 h at room temperature. Eyes were embedded in paraffin and sectioned (5 μm thick) followed by their

staining with HE. Histological slides were imaged with a ZEISS Axio Scan.Z1 slide scanner (Carl Zeiss Microscopy GmBH, Jena, Germany). The data were processed using Zeiss-Zen 3.2 software (blue edition).

## Retinal function

The effects of JC3 and JC4 on retinal function in $Abca4^{-/-}Rdh8^{-/-}$ mice injured with bright light, $Rho^{P23H/+}$ and $Rho^{P23H/P23H}$ mice, and WT C57BL/6J mice were examined by using ERG. Before ERG measurements, mice were anesthetized with a cocktail of 20 mg/ml ketamine and 1.75 mg/ml xylazine at 4 μl/g b.w., and pupils were dilated with 1% tropicamide. Scotopic and photopic ERGs were recorded for both eyes of each mouse using a Celeris rodent ERG system and Espion Dyagnosys software Version 6 (Dyagnosys, LLC, Lowell, Massachusetts, USA). The ERG data were processed for each condition and presented as mean and standard deviation (SD) for both a-wave and b-wave amplitudes. Each experimental group contained at least $n = 5$ mice.

## Detection, quantification, and pharmacokinetics of JC3 and JC4 in mouse eyes

High-performance liquid chromatography (HPLC) coupled with mass spectrometry (MS) was used to examine the eye targeting of JC3 and JC4 compounds. Compounds were administered i.p. to 5-week-old WT C57BL/6J mice. Eyes were collected from euthanized mice at 0.5, 2, 4, 8, and 16 h after compound administration ($n = 4$ mice/time point). Two eyes from each treatment group were pooled together and homogenized on ice in 1 ml of a solution containing acetonitrile/methanol/water at 50:40:10 in the presence of 100 pmol of an internal standard (JC3 or JC4, respectively). These homogenates were mixed with 1,000 μl of hexane and agitated for 2 min followed by centrifugation at 2,200×g for 5 min. The hydrophobic phase was discarded, while the polar phase was collected and dried in a Savant Speedvac concentrator (Thermo Fisher Scientific). The compounds were dissolved in 300 μl of acetonitrile, and 100 μl was injected into an HPLC system. The compounds were separated on the X-Bridge BEH C4 column 2.1 × 50 mm (Waters) using a linear gradient of acetonitrile in water (30% to 100%) for 15 min at a flow rate of 0.3 ml/min. MS-based detection and quantification of the compounds were performed with an LTQ linear ion trap mass spectrometer (Thermo Fisher Scientific) equipped with an electrospray ionization interface operated in the positive ionization mode. Each compound was used to determine the ionization parameters and to tune the instrument. The compounds were detected in the selected reaction-monitoring mode using the following ion transition 314.2 → 238.2 for JC3 and JC4 298.2 → 269.8 for JC4. A calibration curve was determined for each compound by calculating the relationship between the areas for ion intensity peaks corresponding to each compound versus their molar ratios in a range of 10 to 1,000 pmol.

## Regeneration of 11-*cis*-retinal in the mouse eye

The effects of JC3 and JC4 on the recovery of 11-*cis*-retinal after photobleaching were evaluated in 4- to 6-week-old dark-adapted WT C57BL/6J mice. Mice were i.p. injected with a single dose of JC3, JC4, or vehicle. Pupils of these mice were dilated by 1% tropicamide after 30 min prior to their exposure to 10,000 lux white light for 10 min. Mice were then reared in the dark room and euthanized at 0, 2, or 24 h post illumination. Eyes were enucleated and stored in the dark at −80°C. To quantify 11-*cis*-retinal, eyes were homogenized in 1 ml of PBS: methanol (1:1, v/v) supplemented with 40 mM hydroxylamine and incubated for 20 min at room temperature in the dark. Retinoids were extracted twice with 2 ml of hexane. The mixture was

centrifuged at 3,220×g for 5 min at 4°C to separate the hexanes from the aqueous layer. From the top hexane layer, 1.8 ml was transferred to a glass vial. These samples were then dried in a Savant speed-vac concentrator (Thermo Fisher, Waltham, Massachusetts, USA) and re-dissolved in 300 μl of hexane, which was then analyzed by normal phase analytical HPLC column (Zorbax SIL 5 μm, 4.6 × 250 mm) with mobile phase hexane/ethyl acetate at a flow rate of 1.4 ml/min. The signals at 325 nm and 360 nm were collected. Three mice were included per time point. Each eye ($n = 6$) was analyzed individually.

## Statistical analyses

Compound cytotoxicity, thermal shift assay, and cAMP quantification experiments were performed in triplicate and repeated. Each compound was tested at different concentrations. Each assay included positive and negative controls. The opsin-ligand binding, pigment regeneration, thermal stability, and $G_t$ activation assays were performed 3 times. The effect of each compound was shown in a dose-dependent or time-dependent manner. The parameters derived from these measurements were shown as an average and SD. One or two-way ANOVA with Turkey's post hoc tests were used for hypothesis testing. All statistical calculations were performed using the Prism GraphPad 7.02 software. Type1 error tolerance for the experiments was established at 5%. $P$ values <0.05 were considered statistically significant. A different person than the experimenter performed the analysis.

## Supporting information

**S1 Fig. Effect of JC3 and JC4 on opsin plasma membrane expression in 661W cells stably expressing rod opsin.** (A) The fluorescence images of cells treated with the JC compounds at a final concentration of 10 μM or 5 μM 9-*cis*-retinal (9*c*R) labeled with the anti-Rho antibody recognizing the N-terminal epitope of this receptor and the Alexa-Fluor 594-conjugated anti-mouse secondary antibody (red) to detect the cell surface expression. The nuclei of the cells were labeled with DAPI (blue). Scale bar, 10 μm. (B) Immunoblot showing the effect of JC3 and JC4 on the expression level of WT and P23H rod opsin in 661W cells stably expressing these receptors. Total cell extracts (25 μg) were loaded and separated using SDS-PAGE gel, followed by transfer to polyvinyl difluoride membrane (PVDF). Rod opsin was detected with the 1D4 anti-Rho antibody detecting the C-terminal epitope. GAPDH was detected with an anti-GAPDH antibody and used as a loading control. PNGaseF-treated samples were deglycosylated for 1 h at room temperature prior to loading onto the gel. The experiment was repeated 3 times. Representative immunoblots are shown. Raw images of immunoblots present in the figure can be found in S1 Raw Images. (C) Quantification of band intensities of mature and immature P23H rod opsin in non-treated cells, treated with 9-*cis*-retinal and 2 JC compounds. The numerical data can be found in S1 Data.
(EPS)

**S2 Fig. Effect of JC3 and JC4 on retina health in homozygous *Rho*^P23H/P23H^ mice.** (A) Labeling of rod and cone photoreceptors in retina cryosections prepared from eyes of *Rho*^P23H/P23H^ mice treated with JC compounds or vehicle collected at postnatal day (P) 21. The expression of rod opsin was detected with 1D4 anti-Rho antibody (red) and cone opsin with PNA (green). Nuclei are stained with DAPI (blue). The retina center and periphery are shown. Scale bar, 25 μm. (B) The number of nuclei rows in the ONL counted in the retina center and periphery. (C) The RT-qPCR gene expression levels of Rho and M cone opsin. (D) The immunoblot analyses of the Rho and M cone opsin protein expression. The representative images are shown. Raw images of triplicate immunoblots can be found in S1 Raw Images. (E) Quantification of

the Rho and M cone opsin protein expression normalized to GAPDH. (F) Retinal function was examined by the ERG. The measurements were performed in $n = 5$ mice per treatment group. Error bars indicate standard deviation (SD). The statistically significant changes ($P$) observed between vehicle-treated and compound-treated mice are indicated in the figure with asterisks. Statistical significance was calculated with the one-way ANOVA and post hoc Turkey's tests. The numerical data can be found in S1 Data.
(EPS)

**S3 Fig. Toxicity of JC3 and JC4 in vivo.** The effect of chronic or acute administration of JC3 and JC4 on body weight was examined in six-week-old WT C57BL/6J mice. (A and B) Body weight examined in male and female mice ($n = 5$ per treatment group) treated with JC3 or JC4 at 10 mg/kg or vehicle delivered via i.p. injection every other day for 2 weeks. (C) Body weight examined every other day upon acute administration via i.p. injection of JC3 or JC4 at 100 mg/kg, or vehicle. (D–F) The effect of JC3 and JC4 on retinal function and structure examined in WT C57BL/6J mouse eye treated at 10 mg/kg for 2 weeks every other day. The control mice were treated with vehicle. (D) The scotopic and photopic ERG responses were measured in $n = 5$ mice per treatment group. (E) The representative in vivo OCT images of the retina. ONL, outer nuclear layer; INL, inner nuclear layer. Scale bar, 100 μm. (F) Quantification of the ONL thickness measured at 0.5 mm from the ONH. The measurements were performed in $n = 5$ mice per treatment group. Error bars indicate standard deviation (SD). No significant differences in the ERG responses and the ONL thickness were observed between the treatment groups. The numerical data can be found in S1 Data.
(EPS)

**S4 Fig. JC3 and JC4 enantiomer docking to selected rod opsin variants.** Binding pockets of JC3- and JC4-bound human rod opsin homology model and variants. The mutated amino acids are shown in magenta.
(TIF)

**S5 Fig. Enantiomer assignment for JC3 compound.** The isolated enantiomers of JC3 were assigned by the comparison of computed and measured circular dichroism spectra. The CD spectra of isomer 1 (solid blue) and isomer 2 (dashed blue) were recorded in acetonitrile, and their molar ellipticities were plotted against the wavelength. For the sake of comparison, the electronic structures of the R (solid red) and S (dashed red) enantiomers of JC3 were modeled using density functional theory in order to simulate their CD spectra. The computed change in ellipticity for each enantiomer is plotted against the wavelength for the sake of comparison for the lowest energy conformer, which corresponds to the major peak in each experimental structure. The coincidence between the major peaks suggests isomer 1 is R and isomer 2 is S. The numerical data can be found in S1 Data.
(TIF)

**S6 Fig. Binding properties of JC3 and JC4 enantiomer.** (A) The binding of JC3 and JC4 enantiomers to rod opsin was determined by an intrinsic Trp fluorescence quenching assay. Compounds were added to opsin membranes at different concentrations (62.5–1,000 nM) and changes in Trp fluorescence intensity at 330 nm were recorded and plotted as a function of the compound concentration. The binding curves were fitted using PRISM GraphPad 7.02 software. The $K_d$ values of each compound were calculated and averaged from triplicates. These values ± standard deviation (SD) are shown in the figure. (B) The change in the plasma membrane levels of WT and selected rod opsin variants upon treatment with a 10 μM JC3 and JC4 enantiomer mix and enantiopures was quantified and compared to the treatment with vehicle.

The experiment was performed in triplicate. The numerical data can be found in S1 Data.
(EPS)

**S7 Fig. The effect of JC3 and JC4 on the conformational stability of rod opsin variants.** The molecular dynamic (MD) simulations were carried out for the JC3- and JC4-bound human rod opsin homology model and variants. (A) The root mean square fluctuation (RMSF). (B) The root mean square deviation (RMSD). The numerical data can be found in S1 Data.
(TIF)

**S8 Fig. The effect of JC3 and JC4 enantiomers on the PSN in P23H rod opsin.** The specific residue interaction clusters are shown in the protein structures and specified in the tables. A PSN analysis [33] carried out based on MD simulations identified clusters of stabilizing interactions within the native rod opsin structure. The binding of either JC3 or JC4 increased the number of interacting residues within the native clusters or generated additional clusters of interacting residues. Thus, JC compounds generally appear to stabilize the native fold by improving internal residue–residue interactions that form in the context of the RP variants.
(TIF)

**S9 Fig. The effect of JC3 and JC4 enantiomers on the PSN in G51R rod opsin.** The specific residue interaction clusters are shown in the protein structures and specified in the tables. A PSN analysis [33] carried out based on MD simulations identified clusters of stabilizing interactions within the native rod opsin structure. The binding of either JC3 or JC4 increased the number of interacting residues within the native clusters or generated additional clusters of interacting residues. Thus, JC compounds generally appear to stabilize the native fold by improving internal residue–residue interactions that form in the context of the RP variants.
(TIF)

**S1 Raw Images. Raw images for WB in Figs 5D, S1B and S2D.**
(PPTX)

**S1 Data. Numerical Data for Figs 3A, 3B, 3C, 3D, 3E, 4A, 4B, 4C, 4D, 5A, 5C, 5E, 6A, 7C, 7E, 7F, 8E, 8F, 9A, 9B, 9D, 9E, 9F, S1C, S2B, S2C, S2E, S2F, S3A–S3D, S3F, S5, S6A, S6B, S7A and S7B.**
(XLSX)

## Acknowledgments

The authors thank the Visual Science Research Center Core at Case Western Reserve University with special gratitude directed to Catherine Doller for assistance with tissue sectioning and HE staining, Dawn Smith for preparing the eye cryosections, and Maryanne Pendergast for help in tissue imaging. The authors also thank the Small Molecule Drug Discovery (SMDD) Core at Case Western Reserve University for help in screening small molecules used in this study. Finally, the authors thank Christiane Hassel and the Indiana University Flow Cytometry Core Facility and the Indiana University Center for Genomics and Bioinformatics for experimental support.

## Author Contributions

**Conceptualization:** Joseph T. Ortega, Beata Jastrzebska.

**Data curation:** Joseph T. Ortega, Jacklyn M. Gallagher, Andrew G. McKee, Yidan Tang, Miguel Carmena-Bargueňo, Maria Azam, Zaiddodine Pashandi, Marcin Golczak, Jens Meiler, Horacio Pérez-Sánchez, Jonathan P. Schlebach, Beata Jastrzebska.

**Formal analysis:** Joseph T. Ortega, Jacklyn M. Gallagher, Andrew G. McKee, Yidan Tang, Miguel Carmena-Bargueňo, Maria Azam, Zaiddodine Pashandi, Marcin Golczak, Jens Meiler, Jonathan P. Schlebach, Beata Jastrzebska.

**Funding acquisition:** Marcin Golczak, Jens Meiler, Horacio Pérez-Sánchez, Jonathan P. Schlebach, Beata Jastrzebska.

**Investigation:** Joseph T. Ortega, Beata Jastrzebska.

**Methodology:** Joseph T. Ortega, Jacklyn M. Gallagher, Yidan Tang, Miguel Carmena-Bargueňo, Marcin Golczak, Jens Meiler, Horacio Pérez-Sánchez, Jonathan P. Schlebach, Beata Jastrzebska.

**Project administration:** Beata Jastrzebska.

**Resources:** Beata Jastrzebska.

**Supervision:** Jens Meiler, Horacio Pérez-Sánchez, Jonathan P. Schlebach, Beata Jastrzebska.

**Validation:** Joseph T. Ortega, Jacklyn M. Gallagher, Andrew G. McKee, Yidan Tang, Miguel Carmena-Bargueňo, Maria Azam, Marcin Golczak, Beata Jastrzebska.

**Visualization:** Joseph T. Ortega, Yidan Tang, Miguel Carmena-Bargueňo, Maria Azam, Marcin Golczak, Beata Jastrzebska.

**Writing – original draft:** Joseph T. Ortega, Jacklyn M. Gallagher, Andrew G. McKee, Yidan Tang, Miguel Carmena-Bargueňo, Maria Azam, Zaiddodine Pashandi, Marcin Golczak, Jens Meiler, Horacio Pérez-Sánchez, Jonathan P. Schlebach, Beata Jastrzebska.

**Writing – review & editing:** Joseph T. Ortega, Beata Jastrzebska.

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
