## [Editor Report · Decision Letter 0]

12 Aug 2024

Dear Dr Jastrzebska, 

Thank you for submitting your manuscript entitled "Discovery of bioavailable non-retinoid pharmacochaperones that suppress the pathogenic effects of misfolded rhodopsin variants" for consideration as a Research Article by PLOS Biology. Please accept my apologies for the delay in getting back to you as we consulted with an academic editor about your submission.

Your manuscript has now been evaluated by the PLOS Biology editorial staff, as well as by an academic editor with relevant expertise, and I am writing to let you know that we would like to send your submission out for external peer review.

Once your full submission is complete, your paper will undergo a series of checks in preparation for peer review. After your manuscript has passed the checks it will be sent out for review. To provide the metadata for your submission, please Login to Editorial Manager (https://www.editorialmanager.com/pbiology) within two working days, i.e. by Aug 14 2024 11:59PM.

Kind regards,

Richard

Richard Hodge, PhD

rhodge@plos.org

PLOS

---

## [Decision Letter · Decision Letter 1]

24 Oct 2024

Dear Dr Jastrzebska,

Thank you for your patience while your manuscript "Discovery of bioavailable non-retinoid pharmacochaperones that suppress the pathogenic effects of misfolded rhodopsin variants" was peer-reviewed at PLOS Biology as a Research Article. Please accept my sincere apologies for the delays that you have experienced during the peer review process. Your manuscript has now been evaluated by the PLOS Biology editors, an Academic Editor with relevant expertise, and by two independent reviewers. 

Based on the reviews, I am pleased to say we are likely to accept this manuscript for publication, provided you satisfactorily address the remaining points raised by the reviewers. In addition, I would be grateful if you could please make sure to address the following data and other policy-related requests that I have provided below (A-D):

(A) We routinely suggest changes to titles to ensure maximum accessibility for a broad, non-specialist readership. In this case, we would suggest a minor edit to the title, as follows. Please ensure you change both the manuscript file and the online submission system, as they need to match for final acceptance:

“Discovery of non-retinoid compounds that suppress the pathogenic effects of misfolded rhodopsin variants in a mouse model of retinitis pigmentosa”

(B) Please include the full name of the IACUC that reviewed and approved the animal care and use protocol. Please also include an approval number.

(C) Thank you for already providing the underlying raw data for the figures in the S1_Data file. However, we note that the following figures may be missing from the file?

Figures 3D, 8E

In addition, we think that Figures 4C and S1C are mislabeled in the Excel file and we would be grateful if you could correct at this stage. 

(D) Per journal policy, if you have generated any custom code during the course of this investigation, please make it available without restrictions. Please ensure that the code is sufficiently well documented and reusable, and that your Data Statement in the Editorial Manager submission system accurately describes where your code can be found.

*Published Peer Review History*

*Press*

Best wishes,

Richard

Richard Hodge, PhD

rhodge@plos.org

Reviewer remarks:

Reviewer #1: The manuscript by Ortega et al. reports on the discovery of novel non-retinoid molecules that can mitigate the pathogenic effects of rhodopsin mutation associated with inherited retinal disease. The work described is scientifically sound and includes virtual screening of potential ligands that bind to rhodopsin at the orthosteric site, and the characterization of two lead compounds that appear to show good stabilizing properties and in vivo promising positive effects in a mouse model of retinitis pigmentosa. The manuscript is well organized into different relevant sections and the work is clearly described and easy to follow. The methods are described to a good level of detail and the results are discussed in the light of other studies conducted previously. 

A main point to highlight is that the authors have analyzed the potential effect of their discovered compounds on a series of rhodopsin mutations although only used the mouse model for the well-studied P23H mutant. In spite of this, the in vivo work has been conducted with the needed controls (also for safety of the compounds) and it adds physiological significance to the reported conclusions. Overall, the study is rigorous and has scientific merit and it adds to the current efforts trying to develop novel therapeutic strategies for the treatment of rhodopsin retinitis pigmentosa.

As indicated, the manuscript reads well and is not needlessly involved, and the results are clearly presented. Only, a minor aspect is that the insertion of the figure captions within the main text distracts the reader. This maybe a requirement of the journal but it is not clear to this reviewer what is the advantage of this format particularly when the figures are at the end of the main text, anyway. 

I have several points for discussion and clarification.

The authors have two main claims, i) the first is that the use of non-retinoid molecules is advantageous due to the potential damaging effects of free retinoids in the retina and ii) that the use of ligands that bind to the orthosteric (retinal) binding site is a good strategy to counteract the deleterious effects of rhodopsin mutations in inherited retinal degenerative diseases. 

The first claim is reasonable and has been widely accepted in the field although some retinoid derivatives are not proven to be toxic to the retinal cells. However, it is true that some of the investigated retinal analogues can have some potential negative effects on the visual photoreceptor cells.

The second claim, although widely spread in the field, can be subject to some discussion.

1. Concerning the use of retinoids, the authors state that:

"However, their therapeutic utility is limited due to their sensitivity to light and chemical reactivity of retinal photo-metabolites" This is true but the binding of a non- light sensitive molecule to the orthosteric binding pocket can also alter the phototransduction of those opsin molecules by inhibiting it. Wouldn't this potentially cause an alteration of the normal phototransduction pathway that may affect retinal physiology? Please comment.

2. Connected to the previous point, what do the authors think about using molecules that can act as allosteric modulators while 11-cis-retinal is bound in the retinal binding pocket. A number of rhodopsin mutations cause moderate or almost negligible misfolding and those could greatly benefit from targeting other regions other than the orthosteric site. This possibility could be discussed and compared to their strategy of targeting the orthosteric binding site. 

3. "Though promising, many of the 100+ clinical Rho variants failed to respond to these compounds". Could the authors provide some references to this statement?

4. Fig. 2B JC3 10 µM + 9cR, should be changed to JC3 10 µM + 9cR 5 µM. 

What can be the explanation that, in Fig. 2D, when the concentration is greater than 1 µM, the Tm has a downward trend?

5. Fig. 5A, these rod opsin ligands also enhanced the plasma membrane expression of total 36 of 123 tested clinical RP variants, including P23H. Can the authors explain why JC3 and JC4 affect the expression of G90V, but in contrast they almost have no influence on G90D?

6. Do the authors have any concern for long term effects of their compounds on retinal function in vivo?

7. The authors have demonstrated the efficacy of their two compounds, JC3 and JC4through by means of both in vitro and in vivo experiments. This is very exciting and if the authors could further discuss, from the perspective of cellular signaling pathways, how these two molecules can modulate their specific mechanisms of action in vivo, it would be even more comprehensive.

Minor points (the lack of line numbers, and even page numbers, is missed and is somehow troublesome: this makes more difficult to indicate the proposed changes):

In the abstract: "…of currently untreatable blinding diseases called retinitis pigmentosa" suggested change: "…of currently untreatable blinding diseases collectively termed retinitis pigmentosa"

Introduction, third line from bottom. It should be better to refer to RHO as the opsin gene and not the rhodopsin gene.

Second page in the introduction:

"Computational aid screening" should be "Computational-aided screening"…

Reviewer #2: The manuscript "Discovery of bioavailable non-retinoid pharmacochaperones that suppress the pathogenic effects of misfolded rhodopsin variants" by Ortega et al. proposes small molecules to correct rhodopsin misfolding. Quite a number of rhodopsins of Type 2 mutations result in the protein misfolding which are pathogenic ones. They may lead to different currently untreatable blinding diseases known as retinitis pigmentosa. There are different approaches to overcome the problem. The authors of the manuscript describe their search for and the study of "drug-like" small molecules that bind to the orthosteric site of rodopsin improving folding and trafficking of the protein to plasma membranes. Moreover, they found two compounds (JC3 and JC4) which reduce the opsin misfolding and light-degeneration of retina in mice models vulnerable to bright light injury. 

In general, the approach, experiments, computer modelling are described clearly in quite a detail. Taking into account the importance of the problem and promising results which might be useful for further optimization of the molecules I would recommend this work for publication in PLOS Biology. 

Nevertheless, minor revision is necessary. First, the mechanism of JC3 and JC4 is not described in detail. This part of the work is important for further rational optimization of the molecules. Second, at the end of the Discussion section the authors write "Additional investigations are needed to determine whether the efficacy of one or both of these compounds can be enhanced through modification of the structure and/ or dosing regimen in vivo.". However, the weaknesses of both compounds are not described explicitly. Therefore, the readers could not understand well what should be improved to reach the goal. 

Minor comments:

The first paragraph of Introduction. A reference at the end of the sentence "Misfolded membrane proteins that are detected by cellular quality control machinery are retained and degraded in the endoplasmic reticulum (ER), which typically results in a loss of function." is desirable.

The next sentence of the Introduction. The word rhodopsin" should be replaced with "opsin".

The first page of Results. Please add to K296 the word "residue". 

The same page. It would be useful to specify the meaning of the used word "stabilize".

Figure 1. It is useful to show as Figure 1A the overall structure of the protein and the position of the orthosteric side in the protein. (Not all readers are familiar with the molecular structure of rhodopsin).

The part of the text following Table 1. It would be useful for some of the readers to add a reference after "…Lipinsky rule of five".

---

## [Editor Report · Decision Letter 2]

7 Nov 2024

Dear Beata,

On behalf of my colleagues and the Academic Editor, Raquel Lieberman, I am pleased to say that we can accept your manuscript for publication, provided you address any remaining formatting and reporting issues. These will be detailed in an email you should receive within 2-3 business days from our colleagues in the journal operations team; no action is required from you until then. Please note that we will not be able to formally accept your manuscript and schedule it for publication until you have completed any requested changes.

PRESS

Best wishes, 

Richard

Richard Hodge, PhD

rhodge@plos.org

PLOS
